# Dynamic oxygen adsorption on single-atomic Ruthenium catalyst with high performance for acidic oxygen evolution reaction

Linlin Cao [1,6], Qiquan Luo [2,6], Jiajia Chen[2], Lan Wang[3], Yue Lin [2], Huijuan Wang[4], Xiaokang Liu[1], Xinyi Shen[1], Wei Zhang[1], Wei Liu[1], Zeming Qi[1], Zheng Jiang [5], Jinlong Yang [2] & Tao Yao [1]*

Achieving active and stable oxygen evolution reaction (OER) in acid media based on single-atom catalysts is highly promising for cost-effective and sustainable energy supply in proton electrolyte membrane electrolyzers. Here, we report an atomically dispersed $Ru_1-N_4$ site anchored on nitrogen-carbon support (Ru-N-C) as an efficient and durable electrocatalyst for acidic OER. The single-atom Ru-N-C catalyst delivers an exceptionally intrinsic activity, reaching a mass activity as high as 3571 A $g_{metal}^{-1}$ and turnover frequency of 3348 $O_2$ $h^{-1}$ with a low overpotential of 267 mV at a current density of 10 mA $cm^{-2}$. The catalyst shows no evident deactivation or decomposition after 30-hour operation in acidic environment. *Operando* synchrotron radiation X-ray absorption spectroscopy and infrared spectroscopy identify the dynamic adsorption of single oxygen atom on Ru site under working potentials, and theoretical calculations demonstrate that the $O-Ru_1-N_4$ site is responsible for the high OER activity and stability.

[1] National Synchrotron Radiation Laboratory, University of Science and Technology of China, Hefei 230029, China. [2] Hefei National Laboratory for Physical Sciences at the Microscale, University of Science and Technology of China, Hefei 230026, China. [3] School of National Defense Science and Technology, Southwest University of Science and Technology, Mianyang 621010, China. [4] Experimental Center of Engineering and Material Science, University of Science and Technology of China, Hefei 230026, China. [5] Shanghai Synchrotron Radiation Facility, Shanghai Advanced Research Institute, Shanghai 201800, China. [6] These authors contributed equally: Linlin Cao, Qiquan Luo. *email: yaot@ustc.edu.cn

Proton exchange membrane water electrolysis (PEMWE) is widely recognized as a promising approach for storing intermittent solar or wind energy into sustainable hydrogen, owing to its high-energy efficiency, high production rate, and purity[1,2]. The water electrolysis including two half reactions: hydrogen evolution reaction (HER) and oxygen evolution reaction (OER). Unfortunately, compared with the minimal energy losses of HER on the cathode side, the OER on the anodic side still suffers from slow reaction kinetics and poor stability in acidic environment, becoming the main bottleneck of practical implementation of PEMWE[3–5]. Despite great efforts been devoted toward exploring various acidic OER catalysts, considerably large overpotential (>300 mV at 10 mA cm$^{-2}$) is usually required to drive reaction[6,7]. Therefore, the development of OER electrocatalyst, that possess both high activity and stability in acidic media, is highly desirable but remains a grand challenge. To this end, an atomic-level understanding on the nature of the active sites during operation conditions and the catalytic reaction pathways is needed for the rational design of targeted OER catalysts[8–10].

Based on the volcano plot of the binding energy of oxygen intermediates, ruthenium (Ru) and iridium (Ir)-based OER catalysts are currently the mainly known materials of possessing reasonable activity and stability in acidic conditions[11–13]. To balance the cost and the activity, Ru is the cheapest platinum group metal and appears to be an ideal choice. However, the reported OER activity and stability of the state-of-art RuO$_2$ electrocatalysts in acidic solution are far inferior to that in alkaline solution[7,14,15]. In RuO$_2$, the onset of oxygen evolution usually coincides with the onset of Ru dissolution[16,17], which is likely related to the oxidative release of the lattice oxygen evolution reaction (LOER)[18]. As a result, the oxygen-coordinated Ru moieties were leached out of the surface layer of RuO$_2$ during OER in oxidative conditions[19]. Hence, a reasonable means to improve the dissolution resistance of Ru-based OER catalysts is anchoring the single-atomic Ru with oxygen-free coordination in a conductive matrix to avoid the contribution of the LOER. Simultaneously, the coordination environment of single-atomic Ru can be tailored to engineer its electronic properties with the optimized oxygen intermediates bonding energy. The so-called metal-nitrogen-carbon-based (M-N-C) single-atom catalysts with the merits of the perfect atom utility have demonstrated excellent electrocatalytic activity, thus are expected to be an ideal strategy to address the above issues[20–23]. Embedding single-atomic Ru into an acid-resistant N-C coordination environment via forming strong Ru-N bond may help to improve the stability of Ru while endowing it with high activity[24]. Moreover, for single-atom catalysts, the structurally homogeneous and well-defined active site can serve more easily as model systems for atomic-level insight into the catalytic reaction mechanism[25–28].

Here, inspired by the intriguing property of M-N-C structure[29–32], we design a single-atomic Ru confined in carbon nitride-derived N-C support (noted as Ru-N-C) as a robust and efficient electrocatalyst for OER in acid media. At the atomic-level, we unambiguously identify the porphyrin-like Ru$_1$-N$_4$ structural configuration, with a mononuclear Ru coordinated with four N atoms. Interestingly, the operando synchrotron radiation X-ray absorption fine structure (XAFS) spectroscopy and Fourier transform infrared (SR-FTIR) spectroscopy reveal the oxygen pre-adsorption on Ru$_1$-N$_4$ site in the form of O-Ru$_1$-N$_4$ under working potential. Theoretical calculations validate that such O-Ru$_1$-N$_4$ site possesses an optimized binding energy of oxygenated intermediates, and thus boosting OER activity. As a result, the obtained Ru-N-C single-atom catalyst exhibits superior performance in acid electrolyte, reaching the current density of 10 mA cm$^{-2}$ by applying substantially low overpotential of

267 mV for continuous operation over 30 h. Moreover, the Ru-N-C displays exceptionally intrinsic activity, delivering a turnover frequency (TOF) of 3348 O$_2$ h$^{-1}$.

## Results

**Characterization of Ru-N-C catalyst.** The single-atom Ru catalyst was prepared via a facile wetness impregnation of Ru precursor ruthenium chloride (RuCl$_3$) into the dispersion of phosphide carbon nitride, followed by pyrolysis reaction under argon atmosphere (see Methods). The as-prepared Ru-N-C catalyst shows a two-dimensional nano-sheet structure with no obvious particles or clusters of Ru species, as shown by high-resolution transmission electron microscopy image (Supplementary Fig. 1). Inductively coupled plasma optical emission spectrometry results reveal that the mass loading of the Ru is 1.0 wt. %. Note that carbon nitride with abundant unsaturated N with rich electron lone pairs is an ideal anchoring site for immobilizing metal ions to achieve atomic dispersion. The atomic dispersion of Ru species was confirmed by the high-angle-annular-dark-field scanning transmission electron microscopy (HAADF-STEM) characterization with sub-Å resolution. As shown in Fig. 1a, b, the bright spots corresponding to single Ru atoms (sizes of the bright spots are ~0.2 nm) were homogeneously distributed across the entire N-C framework in Ru-N-C[33,34]. Moreover, the intensity profiles along with the direction of X–Y in Fig. 1b uncover that the smallest separated distance of Ru atoms is at least ca. 0.5 nm (Fig. 1c), exceeding the Ru-effective atomic radius, verifying the atomically dispersed Ru on the supports. X-ray diffraction (XRD) results (Supplementary Fig. 2) further confirm the absence of Ru particles or clusters in Ru-N-C[35,36]. Subsequent composition analysis by energy dispersive spectroscopy (EDS) confirms that the obtained Ru-N-C possesses a uniform, uncorrelated spatial distribution of Ru elements throughout the entire sample (Supplementary Fig. 3).

Ru K-edge XAFS measurements were conducted to investigate the Ru local environment at atomic-level. As shown in Fig. 1d, the Fourier transform (FT) of the extended XAFS (EXAFS) curve of Ru-N-C catalyst shows only one dominant peak at 1.5 Å, assigned to the nearest shell coordination of Ru-N/C bond, without the appearance of the Ru–Ru peaks at ~2.3 and 3.1 Å in the FT curves of Ru foil and RuO$_2$, respectively. This precludes the aggregation of Ru-related oxides and clusters, in agreement with the above electron microscopy results. Furthermore, the formation of Ru-N coordination can be confirmed by the soft X-ray absorption spectroscopy at N K-edge and C K-edge. Significant variations in the peaks' intensity can be observed in the N K-edge spectra for the Ru-N-C, whereas no obvious change can be found in C K-edge spectra (Fig. 1e and Supplementary Fig. 4), indicating the strong interaction between N and Ru atoms[37]. Also, the Cl species in the precursor can be reduced and washed out after pyrolysis and calcination, as demonstrated by ion chromatography and XAFS results in Supplementary Fig. 5. It also should be mentioned that the formation of Ru-O-C structures in the as-obtained Ru-N-C catalyst can be excluded, as confirmed by the O K-edge XAS and O 1s XPS measurements (Supplementary Fig. 6). Meanwhile, the P XPS and $^{31}$P solid-state NMR results further demonstrate that partial C atoms have been replaced by P atom, but does not affect the framework (Supplementary Fig. 7). The main purpose of phosphorization is only to improve electrical conductivity of catalyst (Supplementary Fig. 8). Accordingly, the EXAFS fitting analyses were performed as shown in Fig. 1f, and the parameters are summarized in Table S1. The best-fitting results for Ru-N-C clearly show that the major coordination peak originated from the four Ru-N coordination in the form of Ru$_1$-N$_4$ configuration with a mean Ru-N bond length of 2.08 Å,

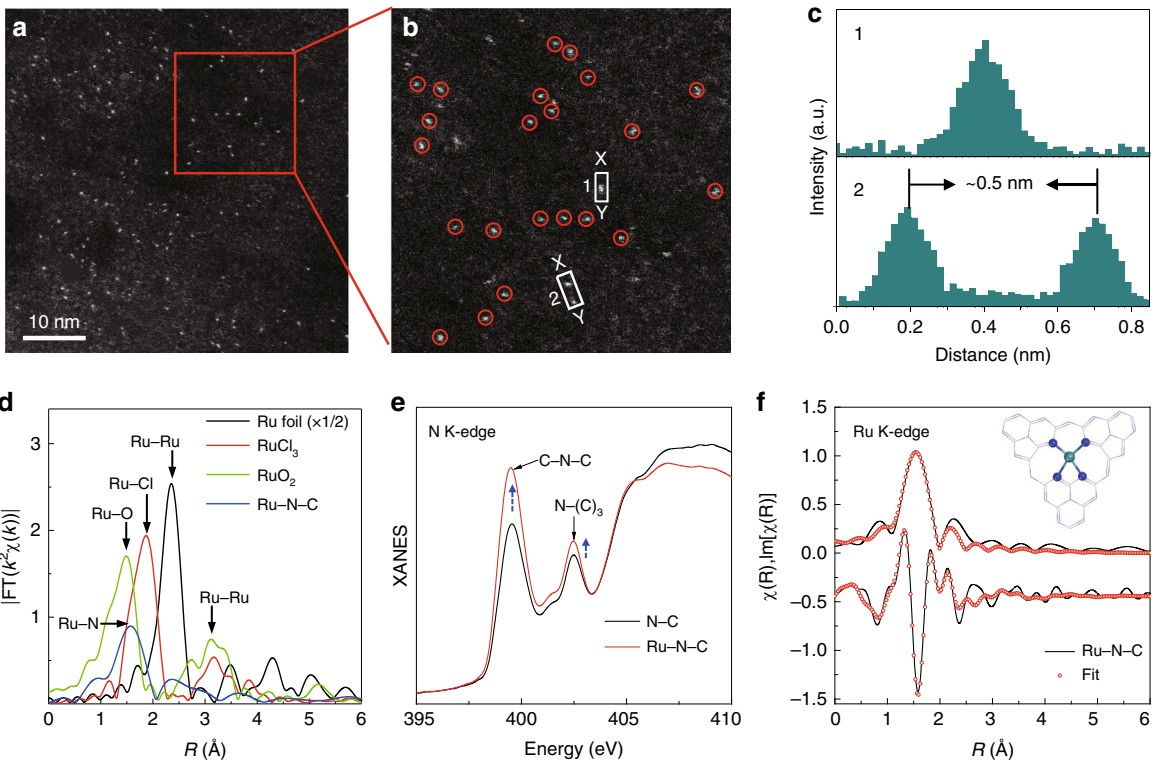

**Fig. 1** Structural characterizations of Ru-N-C. **a** Representative HAADF-STEM image of Ru-N-C catalyst. **b** Magnified HAADF-STEM image of Ru-N-C form the highlighted area of (**a**), showing that the only bright dots corresponding to isolated Ru atoms exist in Ru-N-C. **c** Intensity profiles along the lines X–Y in (**b**). **d** The Ru k-edge $k^2$-weighted Fourier transform spectra for Ru foil, RuCl$_3$, RuO$_2$, and Ru-N-C. **e** N K-edge XANES of pristine N-C and Ru-N-C catalysts. **f** The R-space curve-fitting of ex situ Ru-N-C. Top and bottom curves are magnitude and imaginary part, respectively. Insert shows the structure of the Ru site in Ru-N-C. The balls in gray, blue, and light green represent C, N, and Ru atoms, respectively

suggesting the successful incorporation of the single Ru atoms into the N cavity. Combining the above results, we can conclude that the Ru atoms are atomically dispersed on the N-C support, via bounding with the adjacent four pyridinic-N atoms.

**OER performance of Ru-N-C catalyst.** The electrocatalytic OER activities of Ru-N-C were evaluated in $O_2$-saturated 0.5 M $H_2SO_4$ electrolyte with catalyst-modified glassy carbon electrodes (GCE), along with the results of pristine N-C and commercial RuO$_2$/C for comparison. Figure 2a displays the linear sweep voltammetry curves of all catalysts. Obviously, the as-obtained Ru-N-C exhibits the best acidic OER activity, requiring the lowest overpotential of just 267 mV and 340 mV to achieve a current density of 10 and 100 mA cm$^{-2}$, respectively, outperforming the RuO$_2$/C (300 mV and 515 mV, respectively). The electrochemically active surface area (ECSA) of Ru-N-C was calculated to be 1.42 cm$^2$ with the roughness factor (RF) of 20, which is twice as high as that of RuO$_2$/C (0.86 cm$^2$ and RF = 12) (Supplementary Fig. 9 and Supplementary Note 1). When normalizing the current density to per ECSA, the specific activity of Ru-N-C still surpasses RuO$_2$/C (Fig. 2b). These results clearly show that the atomic Ru coordinated with N is responsible for the high OER activity.

Next, the intrinsic activity of single Ru site was assessed by calculated the TOF (Supplementary Note 2). Notable, the Ru-N-C catalyst displays extremely high intrinsic activity with TOF values up to 3348 $O_2$ h$^{-1}$ and 13392 $O_2$ h$^{-1}$ at the overpotential of 267 and 300 mV (Fig. 2c, Supplementary Fig. 10a), respectively, 394 and 503 times relative to those for RuO$_2$/C (8.5 and 26.6 $O_2$ h$^{-1}$, respectively). Meanwhile, the mass activities of the Ru-N-C are as high as 3571 and 14284 A g$_{metal}$$^{-1}$ at the overpotential of 267 and 300 mV (Fig. 2c, Supplementary Fig. 10b and Supplementary

Note 3), respectively, which are 322 and 410 times relative to those of RuO$_2$/C (11.1 and 34.8 A g$_{metal}$$^{-1}$ at 267 and 300 mV, respectively). Moreover, for Ru-N-C, a smaller Tafel slope of 52.6 mV dec$^{-1}$ and interfacial charge-transfer resistance of 178 Ω were obtained (Fig. 2d and Supplementary Fig. 11), suggesting the faster OER kinetics and electron transfer occurred on single-atomic Ru site. In summary, the performance of the developed Ru-N-C is comparable or superior to other excellent OER catalysts reported to date, such as IrO$_x$/SrIrO$_3$[38], in acidic media (Supplementary Table 3). Compared with other best catalysts those usually need more-precious metals to deliver activity and reasonable stability in acidic electrolyte, our catalyst is much cheaper, only containing C, N, and Ru (low to 1.0 wt.%).

The stability of the catalyst is another essential figure-of-merit for real application, especially in acidic conditions. The stability of the Ru-N-C was assessed via chronoamperometry method at the applied 1.5 V vs. revised hydrogen electrode (RHE) potential. There is a slight degradation (~5 %) over the 30-hour operation (Fig. 2e), in consistent with the results of 1000th cyclic voltammogram (CV) cycles (Fig. 2a). Furthermore, the excellent structural stability of Ru-N-C can also be verified by detecting the dissolution rate of Ru in acidic solution (Fig. 2e). It can be found that the Ru dissolution ratio of as low as 5% can be detected within 30-h operation for Ru-N-C catalyst. Moreover, the morphology and structure remain nearly unchanged after long-time electrolysis, as demonstrated by TEM, XRD, and XAFS results (Supplementary Fig. 12).

Inspired by the excellent acidic OER performance, we conducted the test of overall water splitting in a two-electrode configuration to directly mimic the PEMWE. The Ru-N-C and commercial Pt/C were used as the anode and cathode,

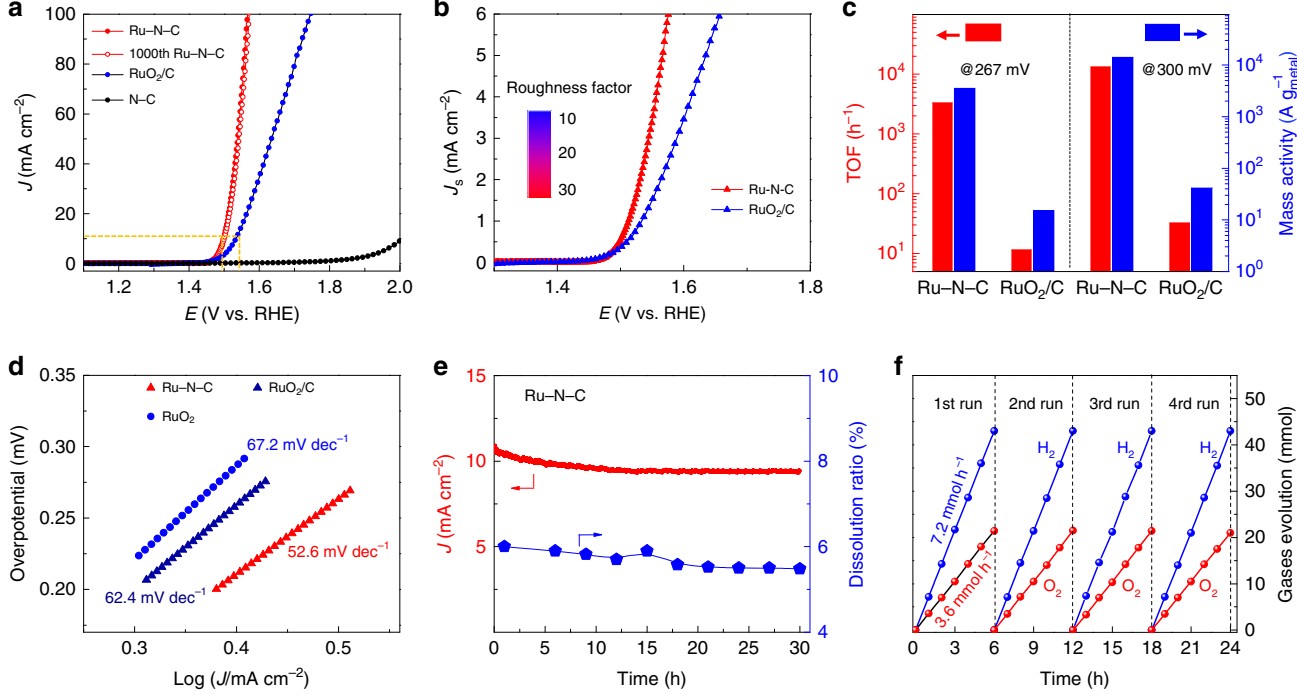

**Fig. 2** Electrochemical OER properties. **a** Electrocatalytic OER performances of the Ru-N-C and commercial $RuO_2$/C in 0.5 M $H_2SO_4$ electrolyte. **b** Normalized linear sweep voltammetry curves to electrochemically active surface area. **c** TOF and mass activities for Ru-N-C and $RuO_2$/C electrocatalysts. **d** Tafel slopes for Ru-N-C and $RuO_2$/C electrocatalysts. **e** Plot of current density and Ru dissolved mass ratio versus time for Ru-N-C at a constant anode voltage of 1.49 V versus RHE in 0.5 M $H_2SO_4$ electrolyte. **f** Time profiles of $O_2$ and $H_2$ evolutions in overall water splitting. All potentials are normalized to RHE

respectively, with the device shown in Supplementary Fig. 13. The amounts of evolved $H_2$ and $O_2$ gas were quantified by means of gas chromatography (Supplementary Fig. 14). Obviously, the Ru-N-C||Pt/C combination exhibits highly full-water splitting activity with mean $O_2$ and $H_2$ evolution rate up to 52 and 104 mmol $h^{-1}$ $cm_g^{-2}$ (where $cm_g^2$ is the electrode area) under the applied voltage of 1.5 V vs. RHE in Fig. 2f, respectively. The quantitative Faradaic gas evolution was at the predicted 2:1 ratio for $H_2$ to $O_2$, within experimental error. Moreover, the electrolyzer can work continuously for 24 h to produce $H_2$ and $O_2$ gaseous products without notable degradation. Considering the exceptional activity and high stability in acidic media, as well as the low cost, the Ru-N-C electrocatalyst may be highly competitive for potential large-scale industrial applications.

**Operando SR-FTIR and XAFS during OER**. To probe the catalytic intermediate, operando SR-FTIR measurements were conducted, as shown in Fig. 3a. At first sight, there is no obvious absorption band can be discerned over the low-vibration frequency region of 900-600 $cm^{-1}$ for the catalyst at 1.2 V vs. RHE or lower potentials. When the higher potentials of 1.5 and 1.6 V vs. RHE were applied, however, a new prominent absorption band appeared at ~764 $cm^{-1}$ in the FTIR spectra (Fig.3b), suggesting the emergence of a crucial intermediate during the OER process. Moreover, when reversing the potential from 1.6 to 1.2 V vs. RHE, the new absorption band gradually disappeared, indicating the reversible adsorption and desorption of intermediate. To clarify the origin of this vibrational absorption band, we calculated via density functional theory (DFT) the different vibrational absorption bands in several possible configurations. As summarized in Supplementary Table 2, we can find that the wavenumber of vibrational absorption band for single oxygen adsorption in the O-$Ru_1$-$N_4$ configuration is quite close to the new peak position observed in the operando SR-FTIR spectra,

suggesting that the intermediate species comes from the single oxygen adsorption (O*).

Operando XAFS measurements using a homemade cell were further conducted to access atomic-level insights into the OER process occurred on single-site Ru-N-C catalyst. The micro-interspace porous carbon clothes were used as working electrode for loading catalysts, so as to assure that nearly all the Ru atoms probed by XAFS are participated into the reaction. The applied working potential was 1.5 V versus RHE in $O_2$-saturated $H_2SO_4$ electrolyte, and the OER has already occurred at this voltage. Figure 3c displays the EXAFS spectra of the catalyst under open circuit and OER condition. The $k^2\chi(k)$ functions show different oscillating shape at the low-$k$ range of 4.0-8.0 $Å^{-1}$ (Supplementary Fig. 15), suggesting the variations of local structures of the single Ru sites. Furthermore, it can be found that, for the FT-EXAFS curve of Ru-N-C catalyst under working conditions (Fig. 3c), the first coordination peak displays low-R shift by 0.07 Å, along with the intensified intensity. As is known, various oxo-containing intermediate species (i.e., O*, OH*, OOH*) would bind on the metal active center during OER process[32,39-41]. Therefore, the variations in FT curves are possibly induced by the adsorption of oxygen-related species. The EXAFS curve-fitting analysis for the first-shell coordination of the catalyst under operando conditions was conducted by considering Ru-N and Ru-O scattering paths, which produce best-fitting quality as shown in Supplementary Figs. 16-17 and Supplementary Table 1. Accordingly, for the catalyst under working condition, the average bond length of Ru-N/O is 2.05 Å, shorter than that of Ru-N (2.08 Å) in the *ex situ* sample. Meanwhile, the fitted Ru-N bond distance is obviously larger than that of Ru-O bond, which we consider is due to the strong interaction and hybridization for Ru-O coordination. This is also in consistent with the structural relaxation in the theoretical calculations. Moreover, the slight shrinkage in Ru-N bonds could further fix Ru atom on the surface, thus avoiding possible

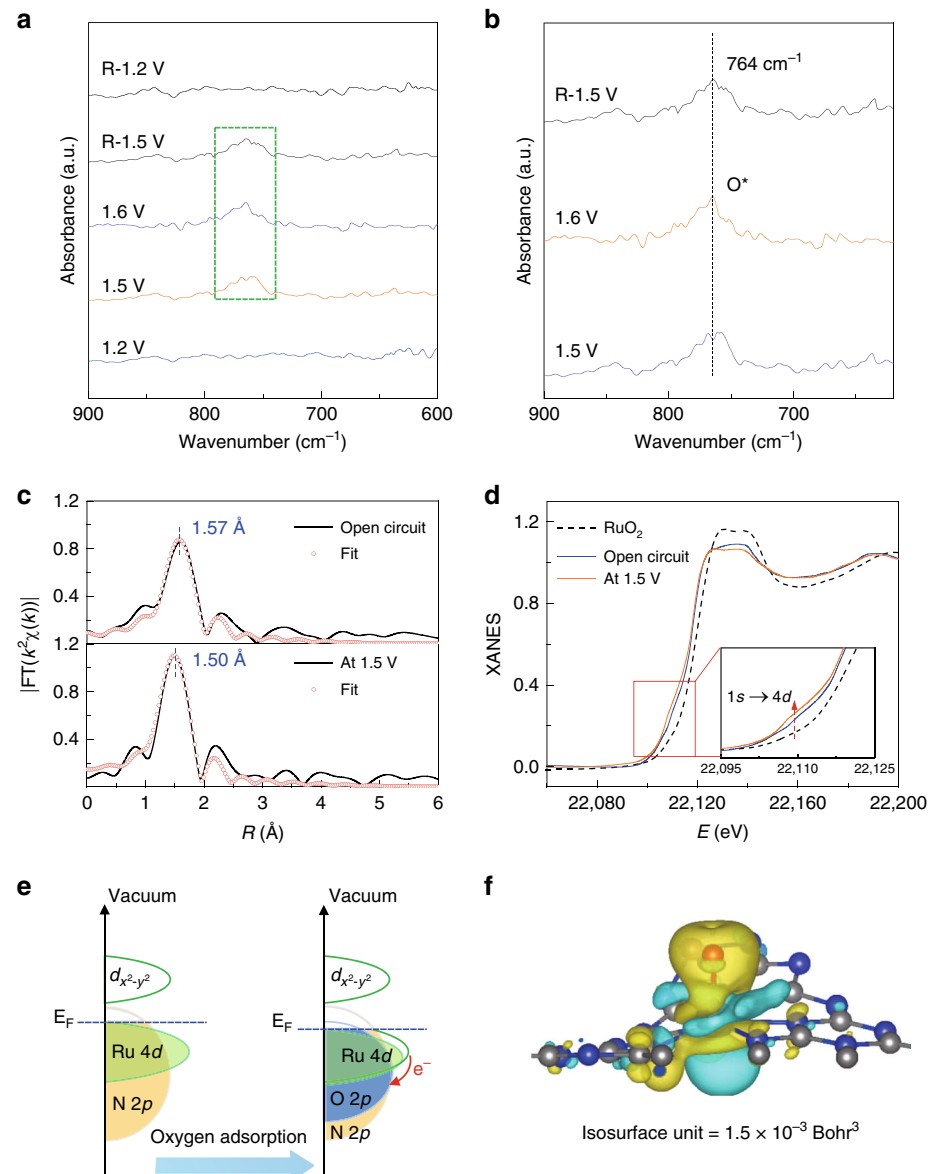

**Fig. 3** Operando SR-FTIR spectroscopy and XAFS measurements. **a** Operando SR-FTIR spectroscopy measurements for Ru-N-C during the acid OER. **b** The enlarged infrared signal at ~764 cm$^{-1}$. All potentials are normalized referred RHE. **c** Operando EXAFS spectra and first-shell fitting curves for Ru-N-C at different applied voltages from the open circuit condition to 1.5 V during OER. **d** Operando XANES spectra for Ru-N-C during OER. Inset: magnified pre-edge XANES region. **e** Schematic illustration of the effect of oxygen adsorption on the electronic structure of Ru-N-C. **f** Electron density difference plot of the O-Ru$_1$-N$_4$. Yellow and light green contours represent the electron accumulation and deletion, respectively. The balls in gray, blue, red, white, and light green represent C, N, O, H, and Ru atoms, respectively. All potentials are normalized to RHE

dissolution when facilitating OER, thus improving the stability of Ru single-atom catalyst.

Accompanied by the evolution of local geometric structure, the changes of the Ru electronic structure can be revealed by the operando X-ray absorption near-edge structure (XANES) results at the Ru K-edge (Fig. 3d). Notably, compared with the highly symmetrical RuO$_2$ octahedron, an apparent pre-edge characteristic feature at around 22115 eV, which arises from the dipole-forbidden but quadrupole-allowed transition of Ru 1$s$ to the unoccupied Ru 4$d$ level, emerges for the catalyst under ex situ and open circuit conditions. The intensity of this shoulder peak is directly proportional to the unoccupied Ru 4$d$-state[15,42]. Hence, the unoccupied Ru 4$d$ level in the ex situ catalyst results from electrons transfers from Ru 4$d$ state to N 2$p$ state via strong Ru-N hybridization in the Ru-N$_4$ site. However, under the working conditions of 1.5 V $vs$ RHE, this pre-edge peak intensifies in

relation to that under open circuit condition. This suggests the distortion of Ru coordination geometry and more electron transfers from Ru 4$d$ state to the nearby atoms under the working state, which may be induced by the additional oxygen adsorption on the Ru, as demonstrated by the operando SR-FTIR and XAFS analysis. Note that although the average valence state of Ru is slightly increased, it still lies between +3 and +4 valence states, as can be reflected by the absorption edges of XANES spectra for Ru-N-C, RuO$_2$, and RuCl$_3$. However, this weak change of Ru oxidation state are hardly to be discerned by the shift of Ru absorption edge, in consistent with the previous reports[43,44]. As schematically shown in Fig. 3e, the oxygen adsorption downshifts the Ru 4$d$ band, resulting in greater covalency of the Ru-N/O bond. This point can be reflected by the more charge donations from Ru obtained by the charge density difference via computational simulation (Fig. 3f). Apparently, compared with ex situ

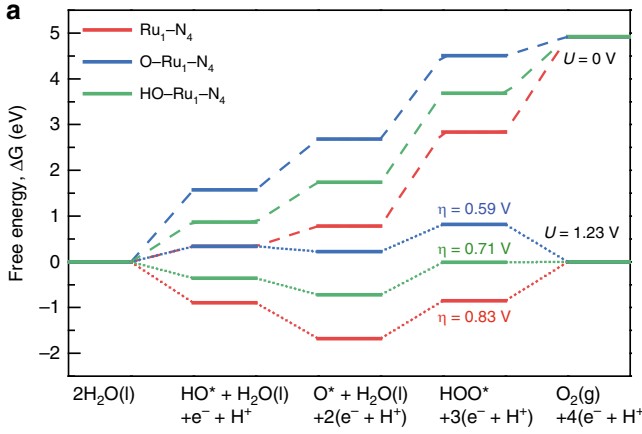

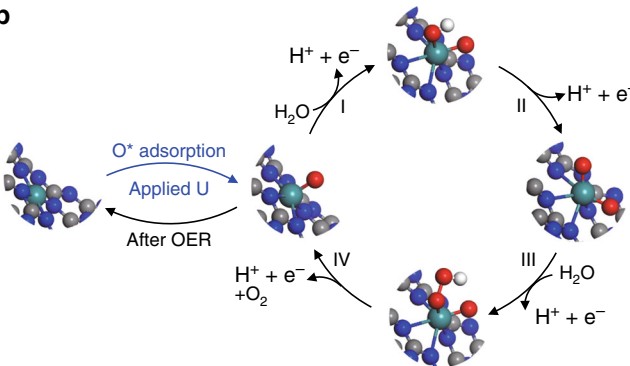

**Fig. 4** Density functional theory calculation for the OER mechanism. **a** Free energy diagram for OER on $Ru_1$-$N_4$ (red line), O-$Ru_1$-$N_4$ (blue line), and HO-$Ru_1$-$N_4$ (green line). **b** Schematic of the whole OER mechanism on Ru-N-C catalyst in the acidic electrolyte. The balls in gray, blue, red, white, and light green represent C, N, O, H, and Ru atoms, respectively. All potentials are normalized to RHE

Ru-N-C, the Ru atoms donates its electrons mainly to adjacent N atoms and the adsorbed O atom through orbital hybridization in O-Ru-N-C. Consequently, we consider the single-atomic Ru has favorable binding energy with OH*, O*and OOH* intermediates, and is responsible for the high OER activity.

**Theoretical insights on OER activity**. To gain further insights into the nature of the high activity of the Ru-N-C and its OER mechanism, DFT calculations were conducted (Supplementary Note 4). For the initial Ru-N-C, the model of single Ru atoms coordinated to four neighboring N atoms was simulated (Supplementary Fig. 18), and the average Ru-N bond distance is obtained as 2.02 Å, which is in good agreement with the EXAFS fitting value. Generally, the conventional OER in an acidic medium involves four concerted proton–electron transfer steps on surface metal sites, generating three different intermediate adsorbates: OH*, O*, and OOH* (the asterisk denotes the adsorption site)[40]. As the Ru site is believed to be the catalytically active center, Fig. 4a and Supplementary Table 4 show the calculated free energy profiles of the OER steps on the Ru site. Obviously, the overpotential-determining step for all the evaluated structural models is the third proton–electron transfer steps of forming OOH* from O*. For the ex situ $Ru_1$-$N_4$, the theoretical overpotential is 0.83 V, significantly larger than the experimental value. It has been reported that, under the OER working potentials, many catalysts would pre-adsorb oxygen species[38,45,46]. Accordingly, we consider that the equilibrium phase of the Ru-N-C catalyst might be covered by oxygen species,

that is, $Ru_1$-$N_4 + H_2O \leftrightarrow H_nO$-$Ru_1$-$N_4 + (2\text{-}n)(H^+ + e^-)$, where $n$ (=0/1) is dependent on the applied potential $U$. Therefore, O or OH pre-adsorption on the Ru-N-C was considered and computed, and the corresponding free energies of O-$Ru_1$-$N_4$ and HO-$Ru_1$-$N_4$ are calculated by the equation: $\Delta G = G_{surf\text{-}ads} - G_{surf} - G_{H_2O} + \frac{2-n}{2}G_{H_2} - (2\text{-}n)eU$. As depicted in Supplementary Fig. 19, the Ru atom is likely covered by a single O atom at 1.5 V versus RHE, forming the O-$Ru_1$-$N_4$ configuration. Hence, the resultant overpotential for O-$Ru_1$-$N_4$ is 0.59 V, much lower than that for the ex situ $Ru_1$-$N_4$ and also the HO-$Ru_1$-$N_4$ (0.71 eV) (Fig. 4a and Supplementary Figs. 20–22). It should be noted that the theoretical overpotential might slightly higher than the experimental value because of the simplified models used in the calculation[32]. Therefore, the theoretical calculations also corroborate our in situ XAFS findings of the oxygen-species adsorption on the Ru. And the decrease of overpotential confirms that the adsorption of single oxygen atom is beneficial for reducing limiting reaction barrier.

The corresponding adsorption configurations of the intermediate species, and the complete OER mechanism, which enlists several steps are summarized in Fig. 4b. The first activation step consists of the adsorption of single oxygen atom on Ru site to form O-$Ru_1$-$N_4$ site, which then reacts with water through nucleophilic attack followed by deprotonation to generate OOH*. A further proton-coupled electron transfer resulted in the release of $O_2$. To understand why O-$Ru_1$-$N_4$ is the facile phase of the OER, the Bader charges of O-$Ru_1$-$N_4$ and ex situ Ru-$N_4$ were investigated accordingly[47]. Ru atom donates 0.88 e and 1.17 e in $Ru_1$-$N_4$ and O-$Ru_1$-$N_4$, respectively, implying the slightly increased average valence state of Ru in O-$Ru_1$-$N_4$ during the OER. Hence, we consider that the O-Ru-$N_4$ with higher Ru oxidation state is the real active site for the high OER activity. Therefore, our simulation results give strong support to the vital responsibility of the isolated Ru site coupled to four N coordination together with single oxygen atom adsorption under operando condition for the superior OER activity of Ru-N-C catalyst.

## Discussion

In summary, we have shown that the single-atom dispersed Ru immobilized on N-C support via forming $Ru_1$-$N_4$ moieties are highly efficient and stable for OER in acidic media. The outstanding OER activity of Ru-N-C catalyst was evidenced by a substantially low overpotential of 267 mV at the current density of 10 mA cm$^{-2}$ for continuous operation of over 30 h in acid operating conditions, which to our knowledge is closed to the best OER electrocatalyst reported in acidic media. More importantly, the combination of in situ XAFS and FTIR allows us to clearly identify the dynamic pre-adsorption of single oxygen atom into the formation of O-$Ru_1$-$N_4$ structure with more charge donations of Ru under operando state. Theoretical calculations further demonstrate that the formed O-$Ru_1$-$N_4$ site under operando state has a low barrier of O-O coupling to form the OOH* intermediate. Hence, this discovery of the structural evolution of active metal site under working conditions can provide a coordination-engineered strategy for designing high-performance acidic OER electrocatalysts towards future PEMWE applications.

## Methods

**Preparation of Ru-N-C**. At first, the graphite carbon nitride (g-$C_3N_4$) nanosheets were prepared via a direct pyrolysis method. In a typical process, 10 g of urea (Sigma-Aldrich, ≥99% purity) were heated to 600 °C with a ramp rate of 5 °C min$^{-1}$ and maintained for 2 h in air atmosphere, and then naturally cooled to room temperature to obtain the yellow powders. The phosphor-decorated carbon nitride (PCN) were synthesized as follows: 300 mg as-obtained g-$C_3N_4$ nanosheets were mixed with 1.5 g NaH$_2$PO$_2$·H$_2$O (Sigma-Aldrich, ≥99% purity, 5:1) sufficiently

through grinding over 30 minutes. Then, the mixture was heated to 300 °C with a heating rate of 2 °C min$^{-1}$ and maintained for 2 h in argon atmosphere. As for Ru-N-C, 200 mg PCN were dispersed in a flask with 60 mL deionized water under continuous sonication for 30 mins and transformed into oil bath. Then, 1.0 mL of RuCl$_3$ (Sigma-Aldrich, ≥99% purity) aqueous solution (10 mg mL$^{-1}$) was dropped into the homogeneous PCN aqueous through microinjection pump and keeping stirring at 70 °C for 5 h. The mixture were further dried via rotary dryer and transferred in the tube furnace and heated at 300 °C for 2 h in argon atmosphere. The obtained powders were ultrasonicated and washed several times with deionized water and ethanol, finally dried under vacuum overnight at 60 °C.

**General characterizations**. The morphology of the catalysts were tested on a JEOL-2100F transmission electron microscopy (TEM) at 200 kV. The field emission scanning electron microscopy (SEM) images and EDS were taken on a Gemini SEM 500 scanning electron microscope. The EDS were conducted in 26FEI Talos F200X at 200 KV. Sub-Å-resolution aberration-corrected HAADF-STEM measurements were conducted on a JEM-ARM 200 F instrument at the accelerating voltage of 200 kV. The powder X-ray diffraction (XRD) patterns were collected on Philips X' pert Pro Super diffractometer with Cu K$_\alpha$ radiation (λ = 1.5418 Å). The concentration of Ru atoms was directly measured by inductively coupled plasma atomic emission spectroscopy (Optima 7300 DV, PerkinElmer, USA). XPS measurements were tested on an ESCALAB MKII instrument equipped with an Mg K$_\alpha$ source (hν = 1253.6 eV). P solid-state NMR spectra were conducted on an AVANCE III 400WB instrument.

**Electrochemical measurements**. The electrochemical measurements were performed using a standard H-type three-electrode cell equipped with an electrochemical workstation (Model CHI760E, CH instruments), which was used to record the electrocatalytic activity in an O$_2$-saturated 0.5 M H$_2$SO$_4$ solution at room temperature. The cell was continuous purged with high-purity O$_2$ for 30 min prior each measurements. Furthermore, the graphite rod was chosen as the counter electrode and Ag/AgCl (3 M KCl) was used as the reference electrode. Approximate 4 mg of the catalysts were ultrasonically dispersed in 1 mL of 3:1 (volume ratio) deionized water and ethanol mix solvent with 20 μL of Nafion solution (5 wt%), then the homogeneous ink (~5 μL) were pipetted out and dropped onto a GCE with a diameter of 3 mm (area: 0.07 cm$^2$) and then fully dried at room temperature. For comparison, commercial RuO$_2$ catalyst (Sigma-Aldrich ≥ 99%) support on carbon black were also prepared by depositing the same mass loading of electrocatalyst on the GCE (mass loading: 0.280 mg cm$^{-2}$) using an identical method. As for OER experiment, anodic linear sweep voltammetry with the 5 mV s$^{-1}$ scan rate with iR drop compensation was carried out form 1.1–2.0 V vs RHE. Electrochemical impedance spectroscopy measurements were tested by applying an AC voltage with 5 mV amplitude in a frequency range from 100 KHz to 100 mHz at overpotential of 300 mV, respectively.

**EECSA and specific activity (js) calculation**. The ECSA for each catalysts were estimated from the electrochemical double-layer capacitance (C$_{dl}$) for the surface of electrocatalyst, which is expected to be linearly proportional to the ECSA. The C$_{dl}$ was determined by measuring the non-Faradic capacitive current associated with double-layer charging form the scan rate dependence of CV. The current response was taking in the potential window for the CV (0–0.1 V vs. Ag/AgCl) with different scan rates (v: 0–100 mV s$^{-1}$)[48]. Moreover, the double-layer charging current (i$_c$) is equal to the product of the scan rate (v) and the electrochemical double-layer capacitance, as given by the (1).

$$i_c = \nu C_{dl} \qquad (1)$$

Thus, a plot of i$_c$ as a function of v obtains a straight line with a slope equal to C$_{dl}$.
The ECSA of the samples are calculated from the C$_{dl}$ according to the (2):

$$ESCA = C_{dl}/C_s \qquad (2)$$

where C$_s$ is the specific capacitance of the certain catalyst of the capacitance of an atomically smooth planar surface of the catalyst per unit area under identical electrolyte conditions. However, the C$_s$ have been measured for various metal electrodes in acidic and typical values reported range between C$_s$ = 0.015–0.110 mF cm$^{-2}$ in H$_2$SO$_4$ solutions. Accordingly, we used the general specific capacitances of C$_s$ = 0.035 mF cm$^{-2}$ in H$_2$SO$_4$ based on typical reported values. All potentials measured were calibrated vs RHE using the following equation:

$$E(RHE) = E(Ag/AgCl) + 0.197V + 0.0592 \times pH \qquad (3)$$

**Operando SR-FTIR spectroscopy measurements**. Operando SR-FTIR spectroscopy measurements were conducted at the beamline BL01B of National Synchrotron Radiation Laboratory (NSRL) via a homemade top-plate cell reflection infrared set-up with a ZnSe crystal as the infrared transmission window with cut-off wavenumber of ~ 625 cm$^{-1}$. This end station was equipped with an FTIR spectrometer (Bruker 66 v/s) with a KBr beam-splitter and various detectors (here a liquid nitrogen cooled mercury cadmium telluride detector was used) coupled with an infrared microscope (Bruker Hyperion 3000) with a ×16 objective, and can

provide infrared spectroscopy measurement with a broad range from 15 to 4000 cm$^{-1}$ with a high spectral resolution of 0.25 cm$^{-1}$.

**Operando XAFS measurements and data analysis**. The Ru K-edge (22117 eV) XAFS spectra were measured at the BL14W1 and 44 A station of Shanghai Synchrotron Radiation Facility (SSRF) and Taiwan Photo Source (TPS). The storage rings of SSRF and TPS were operated at 3.5 and 3.0 GeV with a maximum electron current of 250 and 500 mA, respectively. The data were collected in fluorescence mode using a seven-element Ge detector. Operando XAFS measurements were conducted with the catalyst-modified carbon cloth as the working electrode and using a home-built cell for operando X-ray investigations. Particularly, the catalysts were dispersed in ethanol with 20 μL of Nafion solution (5 wt%, Sigma-Aldrich Co. Ltd.), and then ultrasonicated for 30 min. The well-distributed ink of catalyst was drop-casted onto carbon cloth, which was taped by kapton film on the back and then as working electrode (~1 cm × 1 cm) to ensure all of the electrocatalyst were reacted with H$_2$SO$_4$ electrolyte at a geometric metal loading ~0.5 mg cm$^{-2}$.

First, the working electrode was tested through fluorescence mode without H$_2$SO$_4$ electrolyte and applied voltage, and then named as ex situ. To further estimate the influence of H$_2$SO$_4$ electrolyte, the working electrode was immersed in 0.5 M H$_2$SO$_4$ at open circuit condition and the XAFS spectra were recorded also by fluorescence mode. To monitor the changes during OER process, the anodic voltage was applied at 1.5 V vs. RHE. During the XAFS measurements, we seriously calibrate the position of absorption edge (E$_0$) using Ru foil, and all the XAFS data were collected during one period of beam time. Moreover, each spectrum was measured three times to assure the repeatability of the data and the positions of E$_0$ are almost the same during the multiple scan.

The acquired EXAFS data were processed according to the standard procedures using the ATHENA module implemented in the IFEFFIT software packages[49]. The k$^2$-weighted χ(k) data in the k-space ranging from 2.5 to 10.5 Å$^{-1}$ were Fourier transformed to real (R) space using a hanning windows (dk = 1.0 Å$^{-1}$) to separate the EXAFS contributions from different coordination shells. To obtain the detailed structural parameters around Ru atom in the as-prepared samples, quantitative curve-fittings were carried out for the Fourier transformed k$^2$χ(k) in the R-space using the ARTEMIS module of IFEFFIT[50]. Effective backscattering amplitudes F(k) and phase shifts Φ(k) of all fitting paths were calculated by the ab initio code FEFF8.0[51]. As for the ex situ and under open circuit for Ru-N-C, the FT curves showed single prominent coordination peak at 1.57 Å assigned to the Ru-N coordination. For these three samples, a k range of 2.5–10.5 Å$^{-1}$ was used and curve fittings were done in the R-space within the R-range of (1.0, 2.1) Å for k$^2$-weighted χ(k) functions. The number of independent points are:

$$N_{ipt} = \frac{2 \times \Delta k \times \Delta R}{\pi} = \frac{2 \times (10.5 - 2.5) \times (2.1 - 1.0)}{\pi} = 6 \qquad (4)$$

However, compared with the ex situ sample, the first coordination peak of the catalyst at 1.5 V shown an obvious lower-R shift to 1.50 Å, which was ascribed to be caused by the coexistence of Ru-O coordination. Therefore, we conducted the two shell structure model of a Ru-N and a Ru-O shell to fit the EXAFS data of the sample at 1.5 V. During curve-fittings, each of the Debye–Waller factors (σ$^2$), coordination numbers (N), interatomic distances (R) and energy shift (ΔE$_0$) was treated as adjustable parameter. Notably, in order to reduce the number of adjustable fitting parameters, the N and ΔE$_0$ of Ru-N$_1$ shells were fixed and equal to the parameters of the ex situ sample.

**Soft X-ray XANES measurements**. The N and C K-edge XANES measurements were performed at the BL10B and BL12B beamlines of NSRL. The bending magnet is connected to the beamline, which is equipped with three gratings covering photon energies from 100 to 1000 eV with an energy resolution of ca. 0.2 eV. The data were further recorded in the total electron yield mode by collecting the sample drain current. The resolving power of the grating was typically E/ΔE = 1000, and the photon flux was 1 × 10$^{-10}$ photons per second.

**Computational methods and models**. The plane-wave periodic DFT method in the Vienna ab initio simulation package was used to carry out this work[52]. The electron ion interaction was described with the projector augmented wave method[53]. The electron exchange and correlation energy was treated within the generalized gradient approximation in the form of the Perdew-Burke-Ernzerhof functional[54]. The van der Waals interactions were described by the empirical correction in Grimme scheme (vdW-D3)[55]. A plane-wave basis set of 500 eV was adopted. The force convergence was set to be lower than 0.02 eV/Å, and the total energy convergence was set to be smaller than 10$^{-5}$ eV. Brillouin zone sampling was employed using a Monkhorst-Pack grid[56]. A single layer periodic slab with (4 × 4) supercell pristine and two N atoms defect g-C$_3$N$_4$ was adopted and their vacuum slabs were all set-up to 15 Å. During the calculation all atoms alongside adsorbed intermediates were fully relaxed. Phosphorous-decorated g-C$_3$N$_4$, was modeled by a graphitic carbon atom of g-C$_3$N$_4$ replaced by a phosphor atom and subsequently the corresponding Ru single-atom doped models were optimized as shown in Supplementary Fig. 18.

**Date availability**
The data that support the findings of this study are available from the corresponding author upon reasonable request.

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

## Acknowledgements

This work was supported by the National Key R&D Program of China (2017YFA0208300, 2017YFA0402800 and 2019YFA0210000), the National Natural Science Foundation of China (Grants No. 21471143, 21533007, 11621063, 21688102, and 21703222) and the Fundamental Research Funds for the Central Universities (KY2310000020), and Youth Innovation Promotion Association CAS (CX2310000091). We would thank NSRL, BSRF, and SSRF for the synchrotron beam time. The calculations have been done on the supercomputing system in the Supercomputing Center of University of Science and Technology of China and the High-performance Computing Platform of Anhui University.

## Author contributions

T.Y. developed the idea and designed experiments. L.L.C., L.W., X.K.L., X.Y.S., W.L. W.Z., Z.M.Q., and Z.J. performed the catalyst synthesis and characterizations, FTIR and XAFS measurement, and electrochemical experiments, collected and analyzed the data. Q.Q.L., J.J.C., and J.L.Y. conducted and discussed the theoretical calculations. L.L.C, Y.L., and H.J.W. performed the aberration-corrected STEM characterization. L.L.C. and T.Y. co-wrote the paper. All authors discussed the results and commented on the manuscript.

## Competing interests

The authors declare no competing interests.
