## [Peer Review File · Nature Communications]

Reviewers' comments:

Reviewer #1 (Remarks to the Author):

In this manuscript, the authors report an atomically dispersed Ru1-N4 site anchored on carbon-nitrogen substrate, which acts as an efficient and robust catalyst for oxygen evolution reaction in acidic condition. The single site of Ru catalyst was confirmed by SEM and X-ray absorption spectroscopy. In situ XAS studies were performed to uncover the intermediate species during the catalytic reactions. While the standard characterization and catalytic performance of the current system seem interesting and may attract a certain group of researchers, I am not convinced that their in situ XAS results can support the conclusion drawn. The specific comments are listed below:

1) Shorter Ru-N was observed in the sample under applying 1.5 V potential than that under open circuit condition. It is not necessary that this is due to the additional coordination of oxygen as the change of Ru-ligand first shell can result from the local geometry change as well as the oxidation state change. Moreover, it is highly possible that the XAFS spectrum of the sample under 1.5 V potential can be adequately fitted using the same mode as the ex-situ sample without including the path for Ru-O. It is too rush to draw a conclusion that the shortened Ru-ligand first shell is due to the coordination of oxygen atom.

2) Similarly, it is too rush to draw the conclusion that the slight change of the preedge feature corresponding to 1s-4d transition under 1.5 V potential with respect to ex situ sample is due to electron transfer from Ru to O atom. As the preedge feature is sensitive the local electronic structure of Ru, any change of oxidation state or distortion of the Ru coordination geometry can be responsible for such small change at preedge feature.

3) In situ XAS studies under more applied potentials as well as under more experimental conditions (e.g. in the absence of one of the catalytic reactants) should be performed to confirm whether the intermediate species is due to oxygen coordination.

4) The discussion session is essentially the conclusion. The authors should either add the discussion section or change it to conclusion.

Overall, as the primary novelty of this manuscript is based on the in situ XAS and the observation of Ru-O intermediate species, I do not recommend publication in Nature Communication based on the current version, where the intermediate species cannot be confirmed from the experiments provided.

Reviewer #2 (Remarks to the Author):

In the present work, the authors reported a single-atom Ru-N-C for acidic OER. Overall, the activity of Ru-N-C is relatively low compared with the recently reported Ru-based catalysts (for examples: doi: /10.1002/aenm.201970048). Furthermore, the single-atom Ru-N-C is not a novel material, which has been reported in many other works (for example, J. Am. Chem. Soc., 2017, 139 (28), 9419–9422). Some detailed comments are as follows.

1. The authors demonstrated the Ru-N-C coordination environment based on Ru and N XAS. However, it is not sufficient to exclude the possibility of the formation of Ru-O bonds, e.g., in the form of Ru-O-C. I suggest the authors to measure the O K-edge XAS as well.

2. If there are Ru-O-C moieties, the origin of the OER activity should be reconsidered, as well as the mechanism.

3. The Ru-N-C has a carbon matrix, which can greatly enhance the conductivity. Therefore, it is unfair to compare the OER performance of Ru-N-C and RuO₂. It is suggested to add carbon additive to RuO₂ when preparing the ink. Based on our experience and reported literature (for example, doi: 10.1038/NCHEM.2874), the activity and stability of RuO₂ will be greatly enhanced when it was

incorporated or mixed with conductive carbon.

4. Phosphor-decorated carbon nitride was selected as a support for Ru single atoms. What is the role of phosphor?

5. Given that RuCl_3 is the Ru precursor, the authors should check the Cl content of catalyst. Cl may disturb the OER.

6. Ru dissolution amount was expressed in concentration ($0.27 \mu\text{g L}^{-1}$). However, as a reader, I cannot obtain any useful information from this value. Please give the ratio between dissolved Ru and initial Ru amount of catalyst instead.

7. Detailed characterization should be performed to check the structure of catalyst after 30 h electrocatalysis.

8. The authors should give more information about g-C₃N₄ and Phosphor-decorated carbon nitride.

Reviewer #3 (Remarks to the Author):

Interesting and well-written article. However, only after some concerns are addressed, this manuscript can be accepted.

(1) In the discussion of Fig. 3(b), the authors claim the intensified pre-peak indicates that more electron transfers from Ru 4d state to the nearby atoms under the working state. Can author indicate which atom? If that is the case (electron transfer out of Ru 4d state), the spectral profile should reflect the shift of Ru absorption edge to high energy. However, this shift is not observed, why? The change in pre-peak intensity may also be due to change of Ru local structure. The authors should confirm these issues.

(2) Justify the statement "the higher oxidation state of Ru in O-Ru-N₄ than in Ru-N₄ contributes to the high OER activity".

(3) Is the in situ XAS result reversible at open circuit potential?

(4) A key limitation of commercial Pt/C in the oxygen evolution reaction (OER) is the high cost and high loading of the anodic electrocatalyst. In order to explore further the effect of Ru-N-C, the commercial Pt/C electrocatalyst were evaluated for the OER under identical conditions. The authors should provide the OER polarization curves and Tafel plots of commercial Pt/C compared to that of Ru-N-C and commercial RuO₂ samples toward OER.

(5) RuO₂ and g-C₃N₄ are not properly indexed in Fig. 2 (SI).

(6) To verify the formation of Ru-N-C, the authors should provide the high-resolution SEM or TEM images of Ru-N-C, which has been carried out to confirm the synthesis of Ru-N-C catalysts.

(7) Explain evidently how Ru atoms dispersed singly throughout N-C framework without forming particles or clusters.

(8) In Figure 3(SI), the Ru elements are noticed in the form of clusters? Justify.

(9) Authors mentioned that 0.146 mL of RuCl_3 aqueous solution (10 mg mL^{-1}) dropped into the phosphor-decorated carbon nitride (PCN) aqueous through microinjection pump and stirred at 70°C for 5 h. Can authors explain the chemical reaction mechanism between Ru and PCN in order to form Ru-N-C?

(10) Authors should include indexed SAED pattern. The SEAD pattern should be changed to the SAED pattern (Supplementary Figure 1)

(11) At 1.5 V (Table 1), the bonding distance and coordination number of Ru-N are higher compared to Ru-O why?

Reply to the reviewers' comments:

At first, we sincerely thank all Reviewers for giving us the valuable and thoughtful comments to improve the quality of this manuscript. According to the Reviewers' suggestions, we have performed a number of additional experiments and calculations in order to make following points more clearly. Below we provide point-by-point replies to the Reviewers' comments.

Reviewer #1 (Remarks to the Author):

In this manuscript, the authors report an atomically dispersed Ru₁-N₄ site anchored on carbon-nitrogen substrate, which acts as an efficient and robust catalyst for oxygen evolution reaction in acidic condition. The single site of Ru catalyst was confirmed by SEM and X-ray absorption spectroscopy. In situ XAS studies were performed to uncover the intermediate species during the catalytic reactions. While the standard characterization and catalytic performance of the current system seem interesting and may attract a certain group of researchers, I am not convinced that their in situ XAS results can support the conclusion drawn. The specific comments are listed below:

Reply: We appreciate the reviewer for your encouraging and constructive comments. To support the dynamic adsorption of single oxygen atom on Ru site under working conditions, in this revised manuscript, we have conducted the *operando* synchrotron radiation Fourier transform infrared (SR-FTIR) spectroscopy in Fig. R1. The recorded infrared peak at approximately 764 cm⁻¹ in the SR-FTIR spectra can be ascribed to the vibration adsorption of metal-adsorbed oxygen (O_{ad}). The *operando* SR-FTIR and SR-XAFS, in combination with DFT calculations, provide complementary evidences to the dynamic formation of key Ru-O covalent bond which would influence charge distribution and the local coordination geometry of single-atom Ru site. The detailed discussions are presented in the following.

Question 1. *Shorter Ru-N was observed in the sample under applying 1.5 V potential than that under open circuit condition. It is not necessary that this is due to the additional coordination of oxygen as the change of Ru-ligand first shell can result from the local geometry change as well as the oxidation state change. Moreover, it is highly possible that the XAFS spectrum of the sample under 1.5 V potential can be adequately fitted using the same mode as the ex-situ sample without including the path for Ru-O. It is too rush to draw a conclusion that the shortened Ru-ligand first shell is due to the*

coordination of oxygen atom.

Reply: We thank the reviewer for presenting this nice question. We agree that the shortened Ru-N bond length in the sample under applying 1.5 V potential may be due to the complex atomic and electronic effects, such as the local geometry change and the oxidation state change. The shortened Ru-N coordination, and the more charge transfer from Ru are the phenomenon we observed by using in-situ XAFS. Hence, we consider that it is not suitable to directly state that the shortened Ru-N bond length is due to the additional coordination of oxygen, and have revised this statement in line 220-222 in this revised manuscript.

Hence, to confirm the oxygen adsorption, we have added the *operando* SR-FTIR spectroscopy that was useful for discerning the intermediate adsorption species during electrochemical reactions. As shown in the SR-FTIR spectra (Fig. R1), firstly, there is no obvious absorption band can be observed over the low-vibration frequency region of 900-600 cm^{-1} from 1.0 to 1.2V versus RHE. When applied potential increased to 1.5 V, however, a new absorption band appeared at around 764 cm^{-1} , implying the emergence of a crucial surface-adsorbed intermediate during the OER process. Moreover, when reversing the potential from 1.6 to 1.0 V, this new absorption band gradually disappeared, indicating the reversible adsorption and desorption process. To clarify the origin of this vibrational absorption band, the frequencies calculations were performed on vibrational absorption bands in several possible configurations. As summarized in Table R1, we can find that the wavenumber of vibrational absorption band for single oxygen adsorption in the “O-Ru₁-N₄” configuration is quite close to the new peak position observed in the *operando* SR-FTIR spectra, suggesting that the intermediate species comes from the single oxygen adsorption.

After the verification of the O* adsorption by the direct evidence of SR-FTIR, we thus conducted EXAFS curve-fitting analysis for the first shell coordination under working conditions by considering Ru-N and Ru-O scattering paths. Furthermore, we try to conduct the fitting using the same mode as the ex-situ sample without including the path for Ru-O, and found that the quality and the R-factor are worse than the previous fit, and the corresponding results are shown in Fig. R2 and Table R2. The Table R1 have been added in Supplementary Table 2.

Finally, the theoretical calculations for the OER mechanism in original Fig. 4a also indicated that the isolated Ru atom coupled to four N coordination with single oxygen atom adsorption (O-Ru₁-N₄) possesses lower free energies than Ru₁-N₄, meaning the

O-Ru₁-N₄ was the most favored configuration for OER.

Fig. R1 and revised Figure 3. (a) *Operando* SR-FTIR spectroscopy measurements for Ru-N-C during the acid OER. (b) The enlarged infrared signal at around 764 cm⁻¹. All potentials are normalized referred RHE. (c) *Operando* EXAFS spectra and first-shell fitting curves for Ru-N-C at different applied voltages from the open-circuit condition to 1.5 V during OER. (d) The *operando* XANES spectra for Ru-N-C during OER. Inset: magnified pre-edge XANES region. (e) Schematic illustration of the effect of oxygen adsorption on the electronic structure of Ru-N-C. (f) Electron density difference plot of the O-Ru₁-N₄. Yellow and light green contours represent the electron accumulation and deletion, respectively. The balls in grey, blue, red, white, and light green represent C, N, O, H and Ru atoms, respectively. All potentials are normalized to RHE.

Table R1| DFT calculations for vibrational absorption band in several possible configurations. The red bold represent the vibrated functional groups.

Surface-adsorbed species	Wavenumber (cm ⁻¹)
O -(Ru ₁ -N ₄)	782.3
O -(Ru ₁ -N ₄)-OH	802.8
O -(Ru ₁ -N ₄)-O	783.9
O -(Ru ₁ -N ₄)-OOH	790.4
O-(Ru ₁ -N ₄)- O	838.0
O-(Ru ₁ -N ₄)- OH	917.7
O-(Ru ₁ -N ₄)- OOH	1327.8

Fig. R2| (a) The tentative R-space curve fitting of Ru-N-C at 1.5 V, using the same mode as the ex-situ sample. Top and bottom curves are magnitude and imaginary part, respectively. (b) The corresponding $k^2\chi(k)$ oscillations. All potentials are normalized to RHE.

Table R2| The corresponding structural parameters extracted from the quantitative EXAFS curve-fitting in Fig. R2. (The fixed parameters are underlined).

Sample	Path	N	R (Å)	σ^2 (10^{-3}Å^2)	ΔE_0 (eV)	R -factor
At 1.5V	Ru-N	4.2	2.06 ± 0.08	3.80	0.18 ± 0.9	0.04

Correspondingly, in this revised manuscript, we have re-structured the statements on the verification of Ru-O intermediate species. The *operando* SR-FTIR results were shown in the new Fig. 3a, b, and the corresponding statements were added firstly before EXAFS fitting in line 191-205: “To probe the catalytic intermediate, *operando* SR-FTIR measurements were conducted as shown in Fig. 3a. At first sight, there is no

obvious absorption band can be discerned over the low-vibration frequency region of 900-600 cm⁻¹ for the catalyst at 1.2 V vs. RHE or lower potentials. When the higher potentials of 1.5 and 1.6 V vs. RHE were applied, however, a new prominent absorption band appeared at around 764 cm⁻¹ in the FTIR spectra (Fig.3b), suggesting the emergence of a crucial intermediate during the OER process. Moreover, when reversing the potential from 1.6 to 1.2 V vs. RHE, the new absorption band gradually disappeared, indicating the reversible adsorption and desorption of intermediate. To clarify the origin of this vibrational absorption band, we calculated via DFT the different vibrational absorption bands in several possible configurations. As summarized in supplementary Table 2, we can find that the wavenumber of vibrational absorption band for single oxygen adsorption in the “O-Ru₁-N₄” configuration is quite close to the new peak position observed in the operando SR-FTIR spectra, suggesting that the intermediate species comes from the single oxygen adsorption (O).*”

Question 2. *Similarly, it is too rush to draw the conclusion that the slight change of the pre-edge feature corresponding to 1s-4d transition under 1.5 V potential with respect to ex situ sample is due to electron transfer from Ru to O atom. As the pre-edge feature is sensitive the local electronic structure of Ru, any change of oxidation state or distortion of the Ru coordination geometry can be responsible for such small change at pre-edge feature.*

Reply: We thank the reviewer for presenting this nice question. As we have replied in the last question, we agree that it is not serious to directly attribute the change of pre-edge feature under 1.5 V potential to the charge transfer from Ru to O atom. Since the pre-edge feature corresponds to 1s-4d transition, the intensified pre-edge peak can reflect more unoccupied Ru 4d level, which is caused by the varied Ru coordination geometry. In this revision, we have added *operando* SR-FTIR measurements on Ru-N-C catalyst during the OER process. The SR-FTIR, SR-XAFS results, combined with DFT calculation, have verified the O* adsorption on Ru single atom, leading to the formation of O-Ru-N₄ asymmetric site. As a result, the coordination geometry of Ru was distorted and accompanied by the intensified pre-edge peak at 1.5V potential in *operando* XANES spectra.

Correspondingly, in line 242-246 of this revised manuscript, we have changed the statements into: “*This suggests the distortion of Ru coordination geometry and more electron transfers from Ru 4d state to the nearby atoms under the working state, which*

may be induced by the additional oxygen adsorption on the Ru, as demonstrated by the operando SR-FTIR and XAFS analysis.”

Question 3. *In situ XAS studies under more applied potentials as well as under more experimental conditions (e.g. in the absence of one of the catalytic reactants) should be performed to confirm whether the intermediate species is due to oxygen coordination.*

Reply: We appreciate the reviewer for these nice suggestions. In this work, we have indeed recorded the *operando* XAFS measurements at two another applied potentials: 1.2 and 1.65 V vs. RHE. As shown in Fig. R3a-c, the XANES and EXAFS spectra at 1.2 V is close to those under open circuit condition, indicating no obvious change for the catalyst at 1.2 V, since this is below the standard potential for water oxidation ($E^0=1.23\text{V}$). Moreover, the XANES and EXAFS spectra at 1.65 V is close to those at 1.5 V, suggesting that the structure of single Ru site doesn't undergo the continuous change. Therefore, from the consideration of the clarity of Fig. 2c, d, we only present representative XAFS results under open circuit condition and 1.5 V. According to your suggestions, we have presented all results in the revised Supplementary Fig. 16.

To further confirm the adsorption of single oxygen, we have tried to apply synchrotron beamtime and conduct *operando* XAFS measurements in 0.5 M HClO₄ solution. Since Ru K-edge lies in high-energy region, the synchrotron beamtime is rather limited. We hence performed measurements under two representative conditions: open circuit and at the applied voltage of 1.65 V vs. RHE. As shown in Fig. R3d-f, the XANES and EXAFS spectra for the catalyst at 1.65 V display the similar evolutions relative to those in H₂SO₄ electrolyte. Therefore, based on the above results, we can basically draw the conclusion that the observed intermediate species is from the oxygen adsorption.

Correspondingly, the comparison of XAFS results at different electrolytes have been added as Supplementary Figure 15: *“It can be found that the XANES and EXAFS spectra for the catalyst at 1.65 V in 0.5 M HClO₄ solution display the similar evolutions, as compared to those in H₂SO₄ electrolyte. This further confirm that the observed intermediate species is from the oxygen adsorption.”*

Fig. R3| (a) *Operando* XANES spectra for Ru-N-C at different applied voltages from open-circuit condition to 1.65 V vs. RHE during OER in H₂SO₄ electrolyte. The corresponding $k^2\chi(k)$ oscillations (b), and Fourier transforms (c) for (a). (d) The Ru K-edge XANES spectra for Ru-N-C at open-circuit condition and at 1.6 V in in 0.5 M HClO₄ electrolytes. The corresponding $k^2\chi(k)$ oscillations (e), and Fourier transforms (f) for (e). All potentials are normalized to RHE.

Question 4. *The discussion session is essentially the conclusion. The authors should either add the discussion section or change it to conclusion.*

Reply: We appreciate the reviewer for this reminding. We have changed the “Discussion” title into “Conclusion” in this revised manuscript (line 301).

Reviewer #2 (Remarks to the Author):

In the present work, the authors reported a single-atom Ru-N-C for acidic OER. Overall, the activity of Ru-N-C is relatively low compared with the recently reported Ru-based catalysts (for examples: doi: /10.1002/aenm.201970048). Furthermore, the single-atom Ru-N-C is not a novel material, which has been reported in many other works (for example, J. Am. Chem. Soc., 2017, 139 (28), 9419-9422). Some detailed comments are as follows.

Reply: We appreciate the reviewer for these comments. We are grateful for the reviewer pointing out some nice work reported the Ru-based materials for OER and also the single-atom Ru-N-C materials. Lotsch *et al.* reported NaRuO₂ nanosheet with high acid OER activity, reaching 10 mA cm⁻² at a low overpotential of 255 mV (Lotsch *et al. Adv. Energy Mater.* 2019, 9, 1803795). Li *et al.* reported single Ru atom anchored on metal-organic frameworks or phosphorus nitride for high-efficiency hydrogenation of quinolone or hydrogen evolution reaction (Li *et al. J. Am. Chem. Soc.* 2017, 139, 9419-9422; Li *et al. Angew. Chem. Int. Ed.* 2018, 57, 9495-9500).

Therefore, we have summarized some exciting work in revised Supplementary Table 3 (Ref. 25). Although our Ru-N-C catalyst does not perform best for acid OER, it remains at the top among the best acid OER catalyst. Meanwhile, our Ru-N-C catalyst only contains single-atom Ru with low content, meaning the low cost and high efficiency. Furthermore, another novel point in this work is the identification of dynamic oxygen adsorption on single Ru sites for efficient acid OER, by using *operando* SR-FTIR and SR-XAFS techniques.

Question 1. *The authors demonstrated the Ru-N-C coordination environment based on Ru and N XAS. However, it is not sufficient to exclude the possibility of the formation of Ru-O bonds, e.g., in the form of Ru-O-C. I suggest the authors to measure the O K-edge XAS as well.*

Reply: Thanks for the reviewer's nice question. According to your suggestions, we performed the O K-edge XANES measurements on the as-obtained Ru-N-C catalyst, along with the RuO₂ as reference. As shown in Fig. R4, the XANES spectrum of RuO₂ reference displays the dipole electron transitions from the core level O 1s into unoccupied O 2p projected states above the Fermi level. Due to hybridization between O 2p and Ru 4p, the XANES spectrum splits into double peaks A1 and A2 located at around 529 and 532 eV, respectively. Whereas the broad peak B, centered at ca. 544 eV, is related to the

transitions into O2p-Ru4sp hybridized bands (Chen *et al. Nanoscale*, 2015, 7, 15450). In contrast, the XANES signal for the ex-situ Ru-N-C is very weak as shown in Fig. R4a, suggesting oxygen contents is quite low. Although the O K-edge XANES spectra of Ru-N-C can be seen after magnification, their shapes are completely different from RuO₂. The metallic oxygen peaks at ca. 529 and 532 eV could be hardly discerned in Fig. R4b, basically ruling out the presence of Ru-O coordination in ex-situ Ru-N-C sample, which is consistent with the XAFS fitting results. We consider that these O XAS signals come from the air adsorbed O₂.

To further exclude the possibility of the formation of Ru-O bonds, we also performed the O 1s XPS measurements on the Ru-N-C and RuO₂. The high-resolution O 1s XPS spectrum of RuO₂ shown in Fig. R4c displays two dominant binding energies at ca. 529.6 and 530.7 eV, which can be assigned to the lattice oxygen. Furthermore, a weak peak was observed at ca. 532 eV corresponding to the air-adsorbed O₂ (Antony *et al. J. Phys. D: Appl. Phys.*, 2013, 46, 155202). In contrast, only one binding energy at ca. 532 eV ascribed to the air-adsorbed O₂ was detected in the O 1s XPS spectrum for Ru-N-C (Fig. R4d), indicating the absence of Ru-O chemical bond in ex-situ Ru-N-C sample. This is in accordance with the O K-edge XANES results. It is worth noting that the presence of adsorbed O₂ is unavoidable in the air, while it does not contribute to or influence the catalyst performance.

Furthermore, from the materials synthesis, g-C₃N₄, with the graphene like framework, contains periodic heptazine units, which can provide rich and uniform nitrogen coordinators to capture metal ions. As a result, it is generally reported the formation of “Metal-N/C” structure, rather than the “Metal-O” coordination (Qiao. *et al. J. Am. Chem. Soc.*, 2017, 139, 3336-3339; Pérez-Ramírez *et al. Nat. Nanotechnol.*, 2018, 13, 702.). Therefore, based on above considerations, the possibility of the formation of Ru-O coordination during the pyrolysis process can be basically excluded in the as-obtained Ru-N-C sample.

Accordingly, we have added the Fig. R4 as Supplementary Figure 6, and added the necessary statements in line 123-125 of the revised manuscript: “*It also should be mentioned that the formation of Ru-O-C structures in the as-obtained Ru-N-C catalyst can be excluded, as confirmed by the O K-edge XAS and O 1s XPS measurements (Supplementary Fig. 6).*”

Fig. R4 | (a), (b) O K-edge XANES for RuO₂, Ru-N-C. (c), (d) O 1s XPS high-resolution spectra for RuO₂ and Ru-N-C.

Question 2. *If there are Ru-O-C moieties, the origin of the OER activity should be reconsidered, as well as the mechanism.*

Reply: Thanks for the reviewer's question. We have made detailed discussions in the last question. The added O K-edge XAS and O 1s XPS experiments can basically rule out the possibility of the formation of Ru-O-C coordination in the as-prepared Ru-N-C (Fig. R4). The multiple *operando* techniques including SR-XAFS and SR-FTIR (Fig. R1) demonstrated the reversible adsorption of O on Ru, leading to the formation of O-Ru-N₄. In the whole acid OER process, the "O-Ru-N₄" moieties were verified as the active site and the corresponding OER mechanism is depicted in Fig. 4b.

Question 3. *The Ru-N-C has a carbon matrix, which can greatly enhance the conductivity. Therefore, it is unfair to compare the OER performance of Ru-N-C and RuO₂. It is suggested to add carbon additive to RuO₂ when preparing the ink. Based on our experience and reported literature (for example, doi: 10.1038/NCHEM.2874), the activity and stability of RuO₂ will be greatly enhanced when it was incorporated or mixed with conductive carbon.*

Reply: Thanks for the reviewer's nice question. According your suggestion, we have mixed carbon with commercial RuO₂ to prepare working electrode (RuO₂/C) with a metal loading of about 20%. Typically, approximate 4 mg mixture of RuO₂ and carbon support (Ketjen EC-300J) was first dispersed in in 1 mL of 3:1 (volume ratio) deionized water and ethanol mixed solvent with 20 μL of Nafion solution (5 wt%) under ultrasonication for 30 min. The homogeneous ink (ca. 5 μL) were pipetted out and dropped onto a glassy carbon electrode (GCE) with a diameter of 3 mm (area: 0.07 cm²) and then fully dried at room temperature. As expected, the RuO₂/C exhibits the enhanced OER activity than the bare RuO₂ (Fig. R5a), requiring a lower overpotential of 300 mV to achieve the current density of 10 mA cm⁻². Moreover, the specific activity and mass activity of RuO₂/C also surpass the RuO₂ (Fig. R5b), which is induced by the enhanced conductivity. Although the carbon support can enhance the conductivity and improve the catalytic activity of RuO₂, the OER performance of RuO₂/C is still quite lower than that of Ru-N-C catalyst.

Correspondingly, we have replaced the OER performance curves of RuO₂ by RuO₂/C in the revised Fig. 2.

Fig. R5 (a) The electrocatalytic OER performance for the Ru-N-C, commercial RuO₂ and RuO₂/C in 0.5 M H₂SO₄ electrolyte. (b) Normalized linear sweep voltammetry curves to electrochemically active surface area. (c) TOF and mass activities for Ru-N-C, commercial RuO₂ and RuO₂/C electrocatalysts. (d) Tafel slopes for Ru-N-C, commercial RuO₂ and RuO₂/C

electrocatalysts. All potentials are normalized to RHE.

Question 4. *Phosphor-decorated carbon nitride was selected as a support for Ru single atoms. What is the role of phosphor?*

Reply: We thank the reviewers for pointing out this nice question. To fully clarify the role of P, we have performed substantial experiments including XPS, NMR, XAS, and electrochemical measurements in this revision. On the basis of these additionally experimental results, we conclude that the main purpose of P doping is only to improve the electrical conductivity of the catalyst. The introduction of P dopants hardly affect the framework of g-C₃N₄, or the Ru adsorption site and final structure. Moreover, the OER occurs on the single Ru site, rather than on the P site. The detailed interpretations are presented below from structural and functional aspects.

1. From structural aspects:

At first, the main effect of Phosphor-decorated carbon nitride is to doping P. To reveal the P doping site, we conducted XPS measurements (the original supplementary Figure 5 and Fig. R6a). The P 2*p* XPS spectrum of the Ru-N-C possesses the main XPS peak at 133.4 eV, which can be further fitted with two contributions from the P-N and P=N coordinations (Qian *et al. J. Mater. Res.* 2011, 18, 2359; Chen *et al. Appl. Surf. Sci.* 2018, 430, 309). Moreover, the ³¹P solid-state Nuclear Magnetic Resonance (NMR) spectrum of P-doped C₃N₄ also indicates the formation of P-N coordination (Fig. R6b). Thus, these results strongly indicate that P atoms mostly substitute C atoms in g-C₃N₄ framework (inset of Fig. R6b). These results are in consistent with the previous reports (Zhang *et al. J. Am. Chem. Soc.* 2010, 132, 6294; Qiao *et al. Energy. Environ. Soc.* 2015, 8, 3708; Angew. Chem. Int. Ed. 2015, 54, 4646; Fu *et al. Angew. Chem. Int. Ed.* 2016, 55, 1830; Shi *et al. J. Mater. Chem. A*, 2015, 3, 3862).

Furthermore, the P doping hardly affect the Ru adsorption, and its doping site is far away from Ru. The single-atom Ru would preferably remain bonded with adjacent pyridinic-N atoms in the void of C₃N₄ matrix, rather than on the P dopant. This can be further confirmed by the comparison of XAFS spectra between the Ru-N-C and the RuP₂ reference in Fig. R7. The XANES spectrum of the Ru-N-C is quite different from that of the RuP₂ reference. The FT curve of RuP₂ reference is characterized by two main peaks at about 1.77 Å and 2.4 Å, corresponding to the nearest Ru-P and Ru-Ru coordination, respectively, which are strikingly different from those of the Ru-N-C sample (Fig. R7c).

On the basis of the above results, we consider that the P doping doesn't affect

framework of $g\text{-C}_3\text{N}_4$, and thus has negligible effect on the Ru adsorption.

Fig. R6 | (a) The high-resolution P 2p XPS spectra for the Ru-N-C. Insert: the schematic model of P doping at C site. (b) ^{31}P solid-state NMR spectra of N-C.

Fig. R7 | (a) Ru K-edge XANES spectra for RuP_2 , RuO_2 and Ru-N-C. (b) $k^2\chi(k)$ oscillations of Ru K-edge EXAFS oscillation functions and (c) the corresponding FT curves.

2. From performance aspects:

The main purpose of P doping is to improve electrical conductivity of catalyst. To verify this point, we added the electrochemical impedance spectroscopy (EIS) measurement. Fig. R8a exhibits the EIS Nyquist curves of the P-doped C_3N_4 and initial $g\text{-C}_3\text{N}_4$ at the voltage of 2.0 V, with the corresponding equivalent electrical circuit shown in the inset, in which R_s and R_t refer to the electrolyte solution resistance and the interfacial charge-transfer resistance, respectively. It can be found that P-doped C_3N_4 shows the smaller R_t value in relation to that of $g\text{-C}_3\text{N}_4$, suggesting that the P doping could significantly improve the conductivity, because substitutional n-type P dopants possess more electrons to enhance the delocalized π bonds of $g\text{-C}_3\text{N}_4$. These results are also in consistent with the previous reports (Qiao *et al. Angew. Chem. Int. Ed.* 2015, 54, 4646; Fu *et al. Angew. Chem. Int. Ed.* 2016, 55, 1830).

Moreover, as expected, the OER activity of the P-doped C_3N_4 is higher than pristine $g\text{-C}_3\text{N}_4$, as shown in Fig. R8b and 8c. This indicates that the phosphidation can effectively improve the electrical conductivity of catalysts, and thereby boost the OER activity. It

should be noted that although the P doping can enhance the OER activity, but it is still far inferior to that Ru-N-C, indicating the real active site is still the single-atom Ru site.

Fig. R8 | (a) EIS of the g-C₃N₄ and the P-doped C₃N₄. Insert: the equivalent circuit diagram and the corresponding impedance data. (b) OER performance of the P-doped C₃N₄ and g-C₃N₄ electrocatalysts in 0.5 M H₂SO₄ electrolyte. (c) Overpotential obtained from OER polarization curves at the current densities of 10 and 30 mA cm⁻².

Correspondingly, we have added the Fig. R6, Fig. R7 and R8 as Supplementary Figure. 7 and Supplementary Figure. 8 in the revised Supplementary Materials, and have added the necessary statements in line 123-126 of the revised manuscript: “Meanwhile, the P XPS and ³¹P solid-state NMR results further demonstrate that partial C atoms have been replaced by P atom, but does not affect the framework (Supplementary Fig. 7). The main purpose of phosphorization is only to improve electrical conductivity of catalyst (Supplementary Fig. 8).”

Question 5. Given that RuCl₃ is the Ru precursor, the authors should check the Cl content of catalyst. Cl may disturb the OER.

Reply: We sincerely thank you for this nice suggestion. We consider two main forms of Cl element that may existed in the catalyst, that is, physical and chemical adsorption or bonding with Ru atoms. The Cl species can be basically excluded from the detailed interpretations as following.

1. From synthetic aspects:

The Ru-Cl coordination can be reduced during pyrolysis at protected atmosphere such as nitrogen and argon, which is validated by the previous reports (Li *et al. J Am. Chem. Soc.* 2017, 139, 9419-9422; Li *et al. Angew. Chem. Int. Ed.* 2018, 57, 9495-9500; Xie *et al. Adv. Mater.* 2016, 28, 2427-2431). Meanwhile, the soluble chloride ions in the RuCl₃ precursor can be mostly washed out after multiple washing during the synthetic process,

as can be supported by the results of ion chromatography (Fig. R9). As expected, the content of Cl in final Ru-N-C sample is quite low compared to that in the samples during the first two synthetic stages, including the initial reactive solution (The mixed RuCl₃ and C-N solution) and the immersed sample (The dried former mixture). This indicates that the RuCl₃ precursor is substantially reduced and washed, and the content of Cl in the Ru-N-C is quite low after calcination.

Fig. R9 | The contents of Cl species. The initial reactive solution represented the RuCl₃ aqueous solution mixed with the homogeneous C-N aqueous at 70 °C for 5 h, while the immersed sample is the dried mixture via rotary dryer.

2. From structural aspects:

On the structural side, the absence of Ru-Cl coordination can be confirmed by the XAFS measurements on RuCl₃ reference for comparison (Fig. R10). Although the XANES of initial immersed sample is closed to that of RuCl₃, the XANES spectrum and EXAFS $k^2\chi(k)$ function of the Ru-N-C exhibits obviously different spectral shapes after pyrolysis. The FT curve of RuCl₃ is characterized by the only one peak at around 1.78 Å, corresponding to the first-shell Ru-Cl coordination, which is quite different from the Ru-N coordination in Ru-N-C. Therefore, based on the above discussion, we can rule out the presence of Cl in as-obtained Ru-N-C catalyst.

Fig. R10 | (a) Ru K-edge XANES spectra for immersed Ru-N-C, Ru-N-C and RuCl₃. (b) $k^2\chi(k)$

oscillations of Ru K-edge EXAFS oscillation functions and (c) the corresponding FT curves.

Correspondingly, we have added the Fig. R9 and Fig. R10 as Supplementary Figure. 5 in the revised supplementary materials, and have added the necessary statements in line 121-123 of the revised manuscript: “Also, the Cl species in the precursor can be reduced and washed out after pyrolysis and calcination, as demonstrated by ion chromatography and XAFS results in Supplementary Fig. 5”.

Question 6. Ru dissolution amount was expressed in concentration ($0.27 \mu\text{g L}^{-1}$). However, as a reader, I cannot obtain any useful information from this value. Please give the ratio between dissolved Ru and initial Ru amount of catalyst instead.

Reply: We appreciate the reviewer’s suggestion. According your suggestion, we have changed the dissolution concentration of Ru into the dissolution ratio of dissolved Ru to initial Ru in the catalyst, as shown in the revised Fig. 3e. As shown below, the mean dissolved mass ratio of Ru is quite lower than 6 wt% under acidic OER condition after a long-time electrolysis. The good stability might arise from the strong Ru-N coordination that prevents migration or aggregation for atomically dispersed Ru.

Correspondingly, in line 173-174, we have revised the statement into “It can be found that the Ru dissolution ratio of as low as ca. 5 % can be detected within 30 h operation for Ru-N-C.”

The revised Fig. 2e. Plot of current density and Ru dissolved mass ratio versus time for Ru-N-C at a constant anode voltage of 1.49 V versus RHE in 0.5 M H_2SO_4 electrolyte.

Question 7. Detailed characterization should be performed to check the structure of catalyst after 30 h electrocatalysis.

Reply: We appreciate the reviewer's suggestion. The stability is an important figure-of-merit for the catalyst application. Thus, according your suggestion, the morphology, size and the structure were tested by TEM and XAFS measurements on Ru-N-C after 30-hour OER. As shown in the TEM images (Fig. R11a and b), Ru-N-C catalyst substantially maintained layered structure with no nanoparticles can be found after a long-term electrolysis. Moreover, the XAFS and XRD results further demonstrated that the Ru in Ru-N-C catalyst remain the atomic dispersion without aggregation into the particles (Fig. R11c, d). These results suggest that the Ru-N-C can stably trigger the acid OER.

Correspondingly, the above results have been added in the revised Supplementary Figure 12. And the statements have been added in line 174-176 of this revision: "Moreover, the morphology and structure remain nearly unchanged after long-time electrolysis, as demonstrated by TEM, XRD and XAFS results (Supplementary Fig. 12)."

Fig. R11 | (a) Low-magnification, (b) high-magnification TEM images, and (c) XRD patterns for the Ru-N-C catalyst after long-time electrolysis. (d) The k^2 -weighted Fourier transform EXAFS spectra of catalyst before and after electrochemical test.

Question 8. The authors should give more information about $g\text{-C}_3\text{N}_4$ and Phosphor-decorated carbon nitride.

Reply: We appreciate the reviewer's suggestions. The g-C₃N₄ possesses the graphene like framework, connects with periodic tri-s-triazine units through interlayer van der Waals interactions and the synthesis method is similar to that reported works (Wang *et al. Nat. Mater.* 2009, 8, 76; Kang *et al. Science* 2015, 347, 970-974). On the other hand, regarding the phosphor-decorated carbon nitride, we have made the detail descriptions for your question#4, and have added Supplementary Figure 7 and the necessary statements in this revised manuscript.

Fig. R12 | (a) The high-resolution N1s and (b) C1s XPS spectra. The TEM images for (c), (e) P-doped C₃N₄ and (d), (f) g-C₃N₄.

Reviewer #3 (Remarks to the Author):

Interesting and well-written article. However, only after some concerns are addressed, this manuscript can be accepted.

Reply: We appreciate the reviewer for your evaluation of this work as “*Interesting and well-written article*”.

Question 1. *In the discussion of Fig. 3(b), the authors claim the intensified pre-peak indicates that more electron transfers from Ru 4d state to the nearby atoms under the working state. Can author indicate which atom? If that is the case (electron transfer out of Ru 4d state), the spectral profile should reflect the shift of Ru absorption edge to high energy. However, this shift is not observed, why? The change in pre-peak intensity may also be due to change of Ru local structure. The authors should confirm these issues.*

Reply: We thank the reviewer for these nice comments and suggestions. In this revision, to verify the oxygen adsorption, we have conducted the *operando* SR-FTIR measurements to confirm the Ru-O intermediate species. The detail SR-FTIR interpretation are presented in the reply for the Question #1 of Reviewer #1. As shown in Fig. R1, a new infrared absorption peak was observed at around 746 cm⁻¹ in SR-FTIR spectra for the Ru-N-C at the potential of 1.5 and 1.6 V, implying the formation of surface-adsorbed O* intermediate during the OER process. Therefore, combining the *operando* SR-FTIR, SR-XAFS, and DFT calculations, we demonstrated the dynamic oxygen adsorption to form “O-Ru-N₄” active site under working potential. Meanwhile, such O_{ad} downshifts the Ru 4d band, resulting in more charge transfer from the Ru 4d state to the adjacent N atoms and the adsorbed O atom via orbital hybridization, as supported by calculated charge distribution in Fig. 3d (initial manuscript). Thus, we could draw the conclusion that oxygen adsorption leads to the distortion of the local structure of Ru site, accompanied by the charge transfer that enhances the pre-edge intensity in XANES spectra.

As for the change of the Ru oxidation state, the Bader charge analysis reveals that the Ru atom donates 0.88 *e* and 1.17 *e* in Ru₁-N₄ and O-Ru₁-N₄, respectively, indicating the higher oxidation state for O-Ru₁-N₄. Although the average valence state of Ru is increased, it still lies between +3 and +4 valence states, as can be reflected by the absorption edges of XANES spectra for Ru-N-C, in relation to those of RuO₂ and RuCl₃. However, it is generally reported that absorption edge position of Ru K-edge XANES is not very sensitive to the change of the valence state of Ru between Ru³⁺ and Ru⁴⁺ (for example,

McKeown *et al. J. Phys. Chem. B* 1999, 104, 4825-4832; Scherson *et al. J. Phys. Chem. B* 2000, 104, 9777-9779). Therefore, these weak changes of oxidation state of Ru have not been discerned by the shift of Ru absorption edge.

Correspondingly, we have added the statements in line 246-250 of this revised manuscript: “*Note that although the average valence state of Ru is slightly increased, it still lies between +3 and +4 valence states, as can be reflected by the absorption edges of XANES spectra for Ru-N-C, RuO₂ and RuCl₃. However, this weak change of Ru oxidation state are hardly to be discerned by the shift of Ru absorption edge, in consistent with the previous reports.*”

Question 2. *Justify the statement “the higher oxidation state of Ru in O-Ru-N₄ than in Ru-N₄ contributes to the high OER activity”.*

Reply: Thanks to the Reviewer for pointing out this nice question. We realize that the meaning of previous statements are confused, misleading the reviewer and readers. Our intention is to declare that the O-Ru-N₄ with higher Ru oxidation state is the real active site, rather than the ex-situ Ru-N₄. This question is related to the last question. First, based on the *operando* SR-FTIR and SR-XAFS results, we have identified the formation crucial intermediate species “O-Ru-N₄”. The Bader charges further demonstrated that the Ru atom donates 0.88 *e* and 1.17 *e* in Ru₁-N₄ and O-Ru₁-N₄, respectively, meaning the slightly increased average valence state of Ru in O-Ru₁-N₄ during the OER. As we have replied for the last question, although the average valence state of Ru is increased, it still lies between +3 and +4 valence states, which is hardly to be discerned in the absorption edge position of Ru K-edge XANES, in consistent with the previous reports. Furthermore, the theoretical calculations demonstrate that the O-Ru₁-N₄ site possesses more favorable OER energetics.

Correspondingly, we have changed the previous statements in line 294-298 of this revised manuscript: “*Ru atom donates 0.88 e and 1.17 e in Ru₁-N₄ and O-Ru₁-N₄, respectively, implying the slightly increased average valence state of Ru in O-Ru₁-N₄ during the OER. Hence, we consider that the O-Ru-N₄ with higher Ru oxidation state is the real active site for the high OER activity.*”

Question 3. *Is the in situ XAS result reversible at open circuit potential?*

Reply: This is a nice question. In this revision, we have added the *operando* XAFS results for the catalyst back to open-circuit condition. As shown in Fig. R13, the XANES and

XAFS spectra for the catalyst back to open-circuit condition is quite close to those for the catalyst at initial open-circuit condition. This indicates the reversible adsorption of oxygen, in consistent with the *operando* SR-FTIR results.

Fig. R13 (a) *Operando* XANES spectra for Ru-N-C at open-circuit condition and return to open-circuit condition. (b) $k^2\chi(k)$ oscillations of Ru K-edge EXAFS analysis and (f) The corresponding k^2 -weighted FT spectra.

Question 4. A key limitation of commercial Pt/C in the oxygen evolution reaction (OER) is the high cost and high loading of the anodic electrocatalyst. In order to explore further the effect of Ru-N-C, the commercial Pt/C electrocatalyst were evaluated for the OER under identical conditions. The authors should provide the OER polarization curves and Tafel plots of commercial Pt/C compared to that of Ru-N-C and commercial RuO₂ samples toward OER.

Reply: We thank the reviewer for these comments and suggestions. According your suggestion, the commercial 20 wt% platinum on Vulcan carbon black catalyst (Pt/C from Macklin Biochemical Co., Ltd) was measurement under the same experimental conditions as the Ru-N-C and 20 wt% RuO₂/C. As shown in Fig. R14, commercial Pt/C shows poor catalytic performance and need a substantially higher overpotential of 410 mV to obtain the current density of 5 mA cm⁻² with the larger Tafel slope of 197.1 mV dec⁻¹, in agreement with the previous studies on Pt/C (Yi, J *et al. Electrochem. Commun.*, 2019, 104, 106469; G. Chen *et al. Catal. Today*, 2001, 61, 341-355; Jiang *et al. Chem. Sci.*, 2016, 7, 1690).

Fig. R14 (a) Electrocatalytic OER performances of the Ru-N-C, RuO₂/C, and Pt/C in 0.5 M H₂SO₄ electrolyte. (b) Tafel slopes for Ru-N-C, RuO₂/C, and Pt/C electrocatalysts.

Question 5. RuO₂ and g-C₃N₄ are not properly indexed in Fig. 2 (SI).

Reply: Thanks for the reviewer's criticism. We have added the XRD results of RuO₂ and g-C₃N₄ in the revised Supplementary Figure. 2. The diffraction peaks are well indexed to the (110), (101), (200) and (211) planes of tetragonal RuO₂ phase (JCPDS No. 43-1072; Lin *et al. Nanoscale*, 2014, 6, 2861). Meanwhile, the (110) and (002) planes are belong to g-C₃N₄ (JCPDS No.87-1526; Wang *et al. Nat. Mater.* 2009, 8, 76-80).

Fig. R15 XRD patterns for Ru-N-C, P-doped C₃N₄, and referenced g-C₃N₄, RuO₂.

Question 6. To verify the formation of Ru-N-C, the authors should provide the high-resolution SEM or TEM images of Ru-N-C, which has been carried out to confirm the synthesis of Ru-N-C catalysts.

Reply: We thank the reviewer for this nice suggestion. According your suggestion, more detail HRTEM and HRSEM images were shown in Fig. R16. The SEM results shows that the Ru-N-C have a sheet structure with thin thickness. No Ru nanoparticles or clusters can be observed in HRTEM image. Moreover, energy-dispersive X-ray (EDX) elemental mapping analysis confirms that Ru species are uniformly dispersed in N-C (Fig. R16c, d).

The atomically dispersed Ru species were identified by multiple characterizations of HAADF-STEM and XAFS techniques. Firstly, the HADDF-STEM images with atomic resolution in Fig. 1a and 1b (initial manuscript) clearly reveal that the Ru atoms were uniformly dispersed on the N-C support. This is consistent with EXAFS results, in which only Ru-N/C or Ru-O coordination was detected, without the appearance of the Ru-Ru bonds. Moreover, the Ru-N coordination were identified by the XANES analysis from C, N and O K-edge XAS results.

Correspondingly, in this revision, we have added the HRTEM and HR-SEM images in Supplementary Figure 1, as shown below.

Fig. R16| (a) Low magnification and (b) high magnification SEM images of Ru-N-C. (c) and (d) SEM-EDS elemental mapping of Ru-N-C catalyst.

Question 7. Explain evidently how Ru atoms dispersed singly throughout N-C framework without forming particles or clusters.

Reply: We are grateful to the reviewer for this nice comment. We would answer this question from the following two aspects. Firstly, g-C₃N₄, with the graphene like framework, connects with periodic tri-*s*-triazine units through interlayer van der Waals interactions (Fig. R17). The layered g-C₃N₄ oppresses a C/N ratio near 0.75, and such high level of pyridine-line nitrogen can provide rich electron lone pairs and are potentially ideal sites for metal inclusion (Chai et al *et al.* Chem. Rev. 2016, *116*, 7159-7329; Wang *et al.* Adv. Mater. 2009, *21*, 1609-1612). This will not only offer more metal coordinators but also provide more precise information for the identification of catalytically active sites.

On the other hand, to the best our knowledge, the direct characterization methods that can confirm the single-atom dispersion are the combination of electron microscope and X-ray spectroscopy. The imaging techniques of electron microscopies, especially HAADF-STEM, can directly obtain the distribution of single atom with sub-angstrom resolution (Zelenay *et al.* Science, 2017, *357*, 497; Zhang *et al.* Nat. Chem., 2011, *3*, 634-641). The X-ray spectroscopy, primarily XAS technique, can give clear evidence on the local atomic structure regarding bond distance and coordination number with high resolutions (± 0.01 Å for bond length and $\pm 5\%$ for coordination number, respectively) (Rehr *et al.* Rev. Mod. Phys., 2000, *72*, 621; Wei *et al.* Phys. Rev. Lett., 2010, *105*, 226405). Combined with HAADF-STEM and XAFS results, we not only verified that the individual Ru atoms uniformly dispersed on the surfaces of catalyst, but also coordinated with N atoms. Therefore, based on the above analysis, we can conclude that the Ru atoms dispersed singly throughout N-C framework via Ru-N bond.

Correspondingly, in line 95-97 of this revised manuscript, we have added the statements: “*Note that carbon nitride with abundant unsaturated N with rich electron lone pairs is an ideal anchoring site for immobilizing metal ions to achieve atomic dispersion.*”

Fig. R17| The schematic illustration of the g-C₃N₄.

Question 8. In Figure 3(SI), the Ru elements are noticed in the form of clusters? Justify.

Reply: We thank the reviewer for this question. We consider that the Supplementary Figure 3 may not be clear enough for you and the readers to discern the single-atom dispersion of Ru. We have re-measured the distance between the nearest the bright dots carefully, and found that the distance is about 0.3-0.5 nm, exceeding the radius of Ru atom (ca. 1.34 Å), verifying the single-atom dispersion of Ru. In order to further confirm this point, we provided more energy-dispersive X-ray spectroscopy (EDS) in a scanning transmission electron microscopy (STEM). As shown in Fig. R18, we can found the uniform distributions of Ru in the catalyst. Correspondingly, we have revised the Supplementary Figure 3.

Fig. R18 | STEM-EDS elemental mapping of Ru-N-C catalyst.

Question 9. Authors mentioned that 0.146 mL of RuCl_3 aqueous solution (10 mg mL^{-1}) dropped into the phosphor-decorated carbon nitride (PCN) aqueous through microinjection pump and stirred at 70°C for 5 h. Can authors explain the chemical reaction mechanism between Ru and PCN in order to form Ru-N-C?

Reply: We thank the reviewer for this suggestion. As replied for your Question #7, the PCN contains a large number of unsaturated pyridine-like N atoms, which offer numerous electron lone pairs to capture metal ions. Therefore, the Ru ions in the solution can be strongly bonded to the N atoms when RuCl_3 aqueous solution was immersed into the phosphor-decorated carbon nitride aqueous through microinjection pump and stirred at 70°C for 5 h. Then, the Ru ions were mostly reduced during the pyrolysis (Li *et al.* *J. Am. Chem. Soc.*, 2017, 139, 9419-9422; Xie *et al.* *Adv. Mater.* 2016, 28, 2427-2431),

forming the Ru-N-C structure. To clearly show this reaction process, we have shown a schematic drawing that briefly illustrates the processing steps in Fig. R19.

Fig. R19 | Schematic illustrations of formation mechanism for Ru-N-C.

Question 10. Authors should include indexed SAED pattern. The SEAD pattern should be changed to the SAED pattern (Supplementary Figure 1)

Reply: Thank you for pointing out this mistake in the caption of Supplementary Figure 1. We have changed the “SEAD” to “SAED” in the revised Supporting Information.

Question 11. At 1.5 V (Table 1), the bonding distance and coordination number of Ru-N are higher compared to Ru-O why?

Reply: This is a nice question. We understand the reviewer’s concern that the Ru-O bond distance ought to be higher than Ru-N one, owing to the slightly small scattering amplitude of N. Here, the coordination number of Ru-N is reasonably higher than that of Ru-O, since the Ru is coordinated with four N atoms from support, while is absorbed single O atom from electrolyte at 1.5 V. As we have replied for the above questions, we have added the *operando* SR-FTIR measurements to confirm the presence of single O adsorption on Ru atom at applied potential during OER. Regarding the shorter bond distance of Ru-O, the position of the first coordination peak in EXAFS spectra obviously shifts towards low-R direction for the catalyst at applied potential. Moreover, we consider that the Ru-O has a strong interaction and hybridization, leading to the short Ru-O bond. The shorter bond distance for Ru-O compared to Ru-N is also in consistent with the structural relaxation in the theoretical calculations.

Correspondingly, we have added the necessary explanations in line 225-228: “Meanwhile, the fitted Ru-N bond distance is obviously larger than that of Ru-O bond, which we consider is due to the strong interaction and hybridization for Ru-O coordination. This is also in consistent with the structural relaxation in the theoretical calculations.”

REVIEWERS' COMMENTS:

Reviewer #1 (Remarks to the Author):

The authors have addressed my concerns in the revised manuscript and I thus recommend the publication in Nature Communications.

Reviewer #2 (Remarks to the Author):

My main concern is still the novelty and significance of this study. The relatively low activity of Ru-N-C catalysts is not appealing. And the Ru single-atom catalyst, has also been reported in many other works (for a new example, Nature communications 10(2019): 1711).

Reviewer #3 (Remarks to the Author):

The authors have properly addressed the comments. Therefore, the paper is recommended for publication in the present form.

Reply to the reviewers' comments:

At first, we are sincerely grateful to all the reviewers for their positive advices for the publication of our manuscript. Below we provide point-by-point replies to the Reviewers' comments.

Reviewer #1 (Remarks to the Author):

The authors have addressed my concerns in the revised manuscript and I thus recommend the publication in Nature Communications.

Reply: We appreciate the reviewer for the positive comments and strong support on the publication of this work.

Reviewer #2 (Remarks to the Author):

My main concern is still the novelty and significance of this study. The relatively low activity of Ru-N-C catalysts is not appealing. And the Ru single-atom catalyst, has also been reported in many other works (for a new example, Nature communications 10(2019): 1711).

Reply: We are grateful for the Reviewer for these comments, and pointing out this nice work reported the single Ru electrocatalyst for OER. Sun *et al.* reported a Ru single-atom electrocatalyst with atomically dispersed Ru anchoring on the surface of the cobalt iron layered double hydroxides (Ru/CoFe-LDHs). Their Ru/CoFe-LDHs electrocatalyst exhibits high OER activity in alkaline media, with a low overpotential of 198 mV at 10 mA cm⁻² (Sun *et al. Nat. Commun.* 2019, 10, 1171). Moreover, the *operando* XAFS results demonstrated that the single Ru atom could keep the high-valence state under the applied overpotential due to the strong synergetic electron coupling. Accordingly, in this revision, we have cited the excellent work in Ref. 28.

To the best of our knowledge, the OER performance of our Ru-N-C catalyst is closed to the best reported catalysts, particularly in the acid condition. More importantly, another novel point should be noted is that, by using synchrotron XAFS and IR spectroscopies, we identified for the first time the dynamic oxygen adsorption on the single Ru sites during acid OER process. Theoretical calculations further prove that such site possesses an optimized binding energy of oxygenated intermediates, and thus boosting OER activity.

Correspondingly, in this revised manuscript, we have re-emphasized the novelty in the conclusion.

Reviewer #3 (Remarks to the Author):

The authors have properly addressed the comments. Therefore, the paper is recommended for publication in the present form.

Reply: We appreciate the reviewer for the positive comments and strong support on the publication of this work.